# Hyperthermia-triggered biomimetic bubble nanomachines

Junbin Gao[1], Hanfeng Qin[1], Fei Wang[1], Lu Liu[1], Hao Tian[1], Hong Wang[1], Shuanghu Wang[2], Juanfeng Ou[1], Yicheng Ye[1], Fei Peng[3] ✉ & Yingfeng Tu [1] ✉

Nanoparticle-based drug delivery systems have gained much attention in the treatment of various malignant tumors during the past decades. However, limited tumor penetration of nanodrugs remains a significant hurdle for effective tumor therapy due to the existing biological barriers of tumoral microenvironment. Inspired by bubble machines, here we report the successful fabrication of biomimetic nanodevices capable of in-situ secreting cell-membrane-derived nanovesicles with smaller sizes under near infrared (NIR) laser irradiation for synergistic photothermal/photodynamic therapy. Porous Au nanocages (AuNC) are loaded with phase transitable perfluorohexane (PFO) and hemoglobin (Hb), followed by oxygen pre-saturation and indocyanine green (ICG) anchored 4T1 tumor cell membrane camouflage. Upon slight laser treatment, the loaded PFO undergoes phase transition due to surface plasmon resonance effect produced by AuNC framework, thus inducing the budding of outer cell membrane coating into small-scale nanovesicles based on the pore size of AuNC. Therefore, the hyperthermia-triggered generation of nanovesicles with smaller size, sufficient oxygen supply and anchored ICG results in enhanced tumor penetration for further self-sufficient oxygen-augmented photodynamic therapy and photothermal therapy. The as-developed biomimetic bubble nanomachines with temperature responsiveness show great promise as a potential nanoplatform for cancer treatment.

Over the past decades, malignancies remain one of the most serious threats to public health globally. Although great efforts have been devoted to the development of anticancer agents for either systemic administration or targeted delivery, and nanodrug-based high-precision therapy, the clinical therapeutic efficacy of tumor remains unsatisfied. The limited efficacy is normally attributed to the existence of various biological barriers as well as intricate intratumoral microenvironment, especially for nanodrugs[1–6]. In general, effective penetration of nanomedicines into the deeper tumor region is an essential prerequisite for efficient tumor therapy[7,8]. When nanodrugs reach the tumor region, several obstacles presented by the abnormal tumor microenvironment, including a heterogeneous vascular network, elevated interstitial fluid pressure, extensive stromal cells and dense extracellular matrix (ECM), can significantly hamper the penetration of nanodrugs to the deeper tumor region[9–11]. It's worth noting that the tumor vascular network is also dynamic and fluctuating depending on the tumor type and stage, leading to the uneven perfusion and permeability[12,13], as well as poor distribution of nanodrugs[14–16]. In another aspect, convective flow also has a strong impact on intratumoral diffusion of nanoparticles that extravasated from tumor arteries[17–20]. More importantly, the dense networks of ECM are comprised of numerous components released by stromal and tumor cells

[1]NMPA Key Laboratory for Research and Evaluation of Drug Metabolism & Guangdong Provincial Key Laboratory of New Drug Screening, School of Pharmaceutical Sciences, Southern Medical University, Guangzhou 510515, China. [2]The Laboratory of Clinical Pharmacy, The Sixth Affiliated Hospital of Wenzhou Medical University, The People's Hospital of Lishui, Lishui 323020, China. [3]School of Materials Science and Engineering, Sun Yat-Sen University, Guangzhou 510275, China. ✉e-mail: pengf26@mail.sysu.edu.cn; tuyingfeng1@smu.edu.cn

for the formation of complex 3D physical barriers of tumor tissue[21–23]. To make matters worse, the holes of ECM are typically small with size range from 20 to 130 nm, thus preventing the convection and diffusion of nanoparticles into the tumor and restricting them to the perivascular areas[24,25].

In general, there are two main strategies for nanodrugs to achieve deep tumor penetration, namely tumor microenvironment remodeling (including tumor vasculature and stromal environment) and rational physicochemical regulation of nanoparticles. For the latter, by adjusting the size, morphology and surface property of nanodrugs, efficient tumor penetration can be manipulated accordingly[26,27]. Normally, large nanoparticles (above 100 nm) are advantageous for high tumor accumulation through the extravasation of a leaky vessels based on enhanced permeation and retention (EPR) effect, but it is difficult for them to penetrate into the inner tumoral site due to the huge diffusion hindrance in the tumor matrix[28]. Conversely, nanoparticles with a smaller size are favorable for deep tumor penetration but commonly suffer from short blood circulation time and limited tumor accumulation due to the rapid clearance[29,30]. Consequently, an intelligent delivery system with a size-morphing capability, that maintains a large initial size for selective extravasation, and then transforms smartly into a smaller size at tumor region, is highly desired to achieve tumor accumulation and facilitate deep tumor penetration.

Herein, we demonstrate the ingenious design of biomimetic bubble nanomachines with the capability of tumor-cell-membrane-derived nanovesicle secretion triggered by NIR laser irradiation for enhanced tumor penetration. As illustrated in Fig. 1a, photothermal AuNC with hollow and porous structure are utilized for phase transitable PFO and Hb loading, and the resulting nanoparticles are pre-saturated with oxygen and cloaked subsequently with ICG anchored 4T1 tumor cell membrane. After injecting via tail vein, our biomimetic nanodevices exhibit efficient tumor accumulation due to the homotypic targeting capability of membrane camouflage. Once NIR laser is applied, in situ temperature increases rapidly based on AuNC with surface plasmon resonance effect, thus resulting in phase transition of the loaded PFO. Therefore, the high inner pressure is capable of extruding the coated cell membrane to form nanovesicles fully with oxygen via the pores of AuNC (Fig. 1b). The formed cell-membrane-derived nanovesicles with smaller size further promote efficient tumor penetration for synergistic photothermal and photodynamic therapy.

## Results and discussion

Seed-mediated growth was first used for the synthesis of AgNC. After the fabrication of single-crystal Ag seeds by utilizing ethylene glycol as solvent and silver trifluoroacetate ($CF_3COOAg$) as a precursor[31], the obtained Ag seeds were then mixed with silver nitrate ($AgNO_3$) at suitable ratios in ethylene glycol to synthesize AgNC. According to TEM and SEM, uniform AgNC with the size around 220 nm were successfully synthesized (Fig. 2a and Supplementary Fig. 1a). After adding increasing amounts of $HAuCl_4$ solution, the resulting AgNC were gradually transformed into Ag-AuNC due to galvanic replacement reaction[32], including the initiation of partially hollow nanostructure composed of Ag/Au alloy and final formation of AuNC with pores in the walls (Fig. 2b, c and Supplementary Fig. 1b, c). It is worth noting that by adjusting the amount of the added $HAuCl_4$[33], both the size and the density of the pores of hollow AuNC could be manipulated, which is important as template for the formation of nanovesicles in the follow-up design. Based on ImageJ analysis, AuNC with increased size were clearly observed and the diameter of the formed pores was around 40–50 nm when 30 µg $HAuCl_4$ was used (Supplementary Fig. 1d–f). In order to further endow the nanoparticles with NIR-triggered phase transition property, perfluorohexane (PFO) with the capability of oxygen dissolution based on van der Waals interaction[34] was then loaded into the cavity of hollow AuNC[35]. As natural protein of red blood cells, hemoglobin (Hb) was also introduced into the hollow

nanostructures as secondary oxygen deliverer, followed by oxygen pre-saturation (Fig. 2d). The resulting PFO and Hb loaded AuNC (AuNC-$PO_2$-Hb) were then cloaked with 4T1 tumor cell membrane pre-anchored with photosensitizer indocyanine green (ICG@CCM[36], here referred as ICG@CCM-AuNC-$PO_2$-Hb, thus endowing the whole system with homotypic tumor targeting and synergistic effect of PTT and PDT. As shown in Fig. 2e, f, cell membrane structure with thickness around 8 nm was clearly observed on the surface of AuNC core. According to UV-vis spectroscopy, the characteristic absorption peak of AuNC at 880 nm based on surface plasmon resonance was blue-shifted to 800 nm after ultrasonication and physical co-extrusion with ICG@CCM (Fig. 2g). Meanwhile, the hydrodynamic size of AuNC-$PO_2$-Hb was around 360 nm, which was slightly larger than that of AuNC core (Fig. 2h). Compared with the negatively charged AuNC (−33 mV, Fig. 2i), the introduction of PFO and Hb led to a positive zeta potential of the resulting AuNC-$PO_2$-Hb (16 mV) probably because of the successful loading of the positive PFO nanoemulsion. After subsequent cell membrane coating, zeta potential of final ICG@CCM-AuNC-$PO_2$-Hb overturned to negative again (−31 mV, Fig. 2i), showing the better biocompatibility of our design. In addition, the characteristic peak of ICG was clearly observed in ICG@CCM-AuNC-$PO_2$-Hb under the emission of 780 nm, indicating the successful ICG anchoring during the cell membrane coating (Fig. 2j). Energy-dispersive X-ray spectroscopy (EDX) was further used to map the elemental presence on our biomimetic ICG@CCM-AuNC-$PO_2$-Hb. Significant Au (from AuNC) and C/N/O (from cell membrane and Hb) enrichment were clearly observed as shown in Fig. 2k–i, especially 12% increasement in atomic percentage of C compared with AuNC (Supplementary Fig. 1g, h), indicating the successful coating of the surrounding organic layer and Hb loading. The loaded Hb and the proteins from 4T1 cell membrane in ICG@CCM-AuNC-$PO_2$-Hb were further evaluated by sodium dodecyl sulfate-polyacrylamide gel electrophoresis (SDS-PAGE) (Supplementary Fig. 2). After cell membrane coating, 4T1 cell membrane proteins were almost preserved, and the loaded Hb (65 kD) were also observed in the protein profiles of both ICG@CCM-AuNC-$PO_2$-Hb and the secreted CCM nanovesicles. The Hb and ICG loading content were then calculated according to the standard curves based on UV-Vis absorbance and fluorescence spectrum (Supplementary Fig. 3a, b), which was around 6.03% and 11.19% respectively. We also tested the stability of the loaded Hb, and the circular dichroism analysis showed that the activity of Hb after loading was not significantly affected (Supplementary Fig. 4a, b).

After confirming the successful fabrication of our ICG@CCM-AuNC-$PO_2$-Hb, photothermal conversion capability was then recorded by a real-time infrared (IR) thermal camera (Fig. 3a). Upon laser irradiation for 6 min (808 nm, 1.0 W cm$^{−2}$), the temperature of the solution containing ICG@CCM-AuNC-$PO_2$-Hb, AuNC-$PO_2$-Hb, AuNC, and CCM-AuNC-$PO_2$-Hb with same Au concentration of 100 µg mL$^{−1}$ exhibited a rapid increase to nearly 50 °C. However, the temperature of PBS used as a control was slightly elevated at only 1.2 °C under the same laser condition. The photothermal conversion efficiency ($\eta$) of ICG@CCM-AuNC-$PO_2$-Hb was also calculated and $\eta$ value was around 26.81 ± 1.3% (see Supplementary Information for details, Supplementary Fig. 5), suggesting the developed system showed great potential as photothermal agent. Furthermore, photostability of ICG@CCM-AuNC-$PO_2$-Hb was investigated as well (Fig. 3b). For the consecutive four cycles, 70 s laser treatment followed by intervals to cool down to room temperature was performed to evaluate the sustainability of photothermal conversion. Only 4 °C decline was observed after four heating-cooling cycles, indicating relative photostability of our ICG@CCM-AuNC-$PO_2$-Hb. According to the previous report[37], the transition temperature of the loaded PFO is around 58 °C, which can be easily gasified by the surface plasmon resonance effect from AuNC under laser irradiation. Therefore, the gaseous PFO is possible to blast from the etched

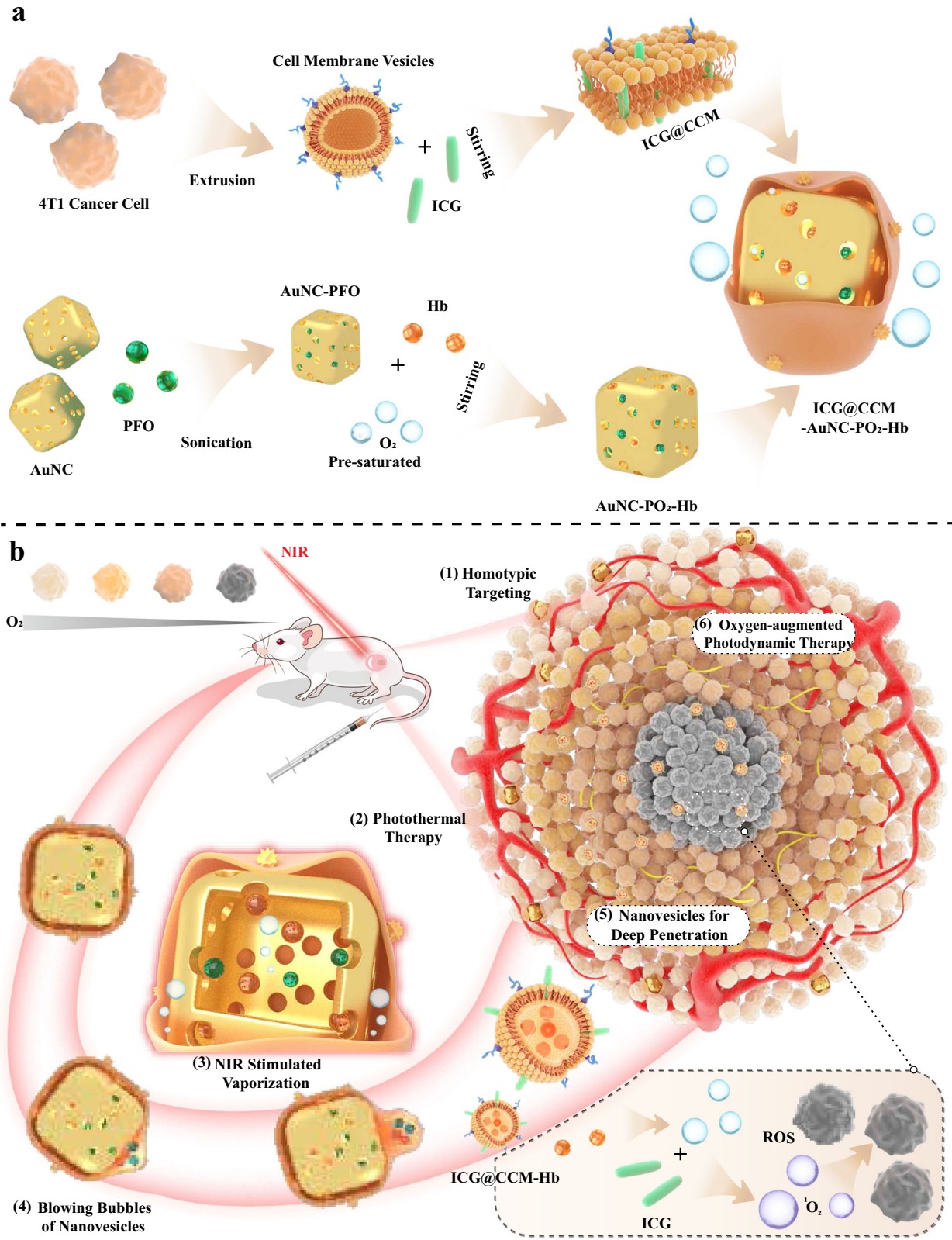

**Nature Communications** | (2023)14:4867

pores of AuNC and further extrude the cloaked cell membrane to form nanovesicles with a smaller scale based on the pore size of AuNC, which is promising for enhanced penetration into the tumor. In addition to the photothermal effect, the photodynamic property of ICG@CCM-AuNC-PO$_2$-Hb was further evaluated by using typical Singlet Oxygen Sensor Green (SOSG) to confirm the $^1O_2$ generation under the same laser condition (808 nm, 1 W cm$^{-2}$, 2 min). As shown

in the Supplementary Fig. 6, ICG@CCM-AuNC-PO$_2$-Hb exhibited great $^1O_2$ production efficacy, which was 2.5-fold higher than that of PBS.

As many reports have proven that PFO has high oxygen-carrying capability[38,39], it is expected that our biomimetic ICG@CCM-AuNC-PO$_2$-Hb could achieve oxygen dissolution and hyperthermia-triggered oxygen release. After pre-saturating with O$_2$, the oxygen loading

**Fig. 1 | Schematic illustration of the fabrication of biomimetic bubble nano-machines with small-scale nanovesicle secretion for synergistic photothermal/photodynamic therapy. a** Design of temperature-responsive biomimetic nano-devices by cloaking 4T1 cell membrane onto PFO/Hb-loaded AuNC with oxygen pre-saturation (ICG@CCM-AuNC-PO$_2$-Hb). The inside abbreviations [AuNC] =Gold nanocage, [PFO] = Perfluorohexane, [ICG] = Indocyanine green, [ROS] = Reactive oxygen species, [Hb] = Hemoglobin. **b** Laser-triggered nanovesicles secretion for enhanced tumor penetration:, (1) Homologous tumor targeting of ICG@CCM-AuNC-PO$_2$-Hb based on the cloaked 4T1 cell membrane; (2) Laser-induced photo-thermal therapy of ICG@CCM-AuNC-PO$_2$-Hb; (3) Hyperthermia trigger the eva-poration of inner PFO under NIR irradiation; (4) The gaseous PFO extrude the cloaked cell membrane for the formation of biomimetic nanovesicles (ICG@CCM-Hb); (5) Penetration of the generated ICG@CCM-Hb with smaller size into deeper tumor region; (6) ICG@CCM-Hb deliver oxygen into deep tumor for self-sufficient oxygen-augmented photodynamic therapy upon second laser irradiation.

capacity of ICG@CCM-AuNC-PO$_2$-Hb was then measured, which was around $0.65 \pm 0.09$ mg per 1 g of ICG@CCM-AuNC-PO$_2$-Hb (Supple-mentary Fig. 7). In order to further monitor the release behavior of O$_2$, a portable oxygen meter was used to record the concentration of the dissolved oxygen in real-time. Notably, upon laser irradiation (808 nm, 1 W cm$^{-2}$), the concentration of the released oxygen from ICG@CCM-AuNC-PO$_2$-Hb was increased sharply to 8.4 mg L$^{-1}$ within 10 min, compared with only 5.8 mg L$^{-1}$ at the same condition without NIR treatment (Fig. 3c). The laser triggered oxygen release was mainly attributed to hyperthermia-induced PFO evaporation under laser irradiation, accompanied by a decline in oxygen solubility in higher temperature in the meantime[40,41]. We supposed that gaseous PFO together with oxygen was capable to produce cell-membrane-based nanovesicles by extruding through the pores from AuNC. In order to confirm the generation of nanovesicles, nanoparticles tracking analy-sis (NTA) was then utilized to monitor the changes in the size and the number of nanoparticles under different laser treatments. As shown in Fig. 3d, the total number of nanoparticles in the group of ICG@CCM-AuNC-PO$_2$-Hb after laser irradiation was 2.1 folds higher than that without laser treatment. Especially, in the size range of nanoparticle less than 100 nm, the particle number increased substantially to 3.3 times (Fig. 3e). For the size range from 100 to 200 nm, the number of the resulting nanoparticles was also increased by 2.2 folds under 0.5 W cm$^{-2}$ when compared with the control group (Fig. 3f), whereas only 1.8 folds increase was observed for the particle's size larger than 200 nm (Fig. 3g). The formed nanovesicles were further collected for DLS measurement and their particle sizes were mainly distributed in the range of 43-120 nm (Supplementary Fig. 8). The above-mentioned results indicated that our smart bubble nanomachines could massively produce nanoparticles with small-scale upon laser irradiation, which would be promising for light-triggered tumor penetration.

TEM was further used to visualize the whole process of nanovesicles formation of ICG@CCM-AuNC-PO$_2$-Hb under NIR irradiation. As shown in Fig. 3h, membrane-derived vesicles were indeed extruded and remained on the surface of AuNC framework. Super-resolution microscope was also utilized to record the dynamic secretion process of our nanodevices in real time. Since certain fluorescent molecules (such as Alexa 647) capable of con-tinuous flashing is suitable for super-resolution imaging (resolu-tion of a-axis and y-axis up to 20 nm), Alexa-647 secondary antibodies were then integrated with the overexpressed CD44 proteins from the cloaked 4T1 cell membrane as fluorescent probe, therefore the changing in cell membrane could be easily mon-itored. In order to reduce the interference from the Brownian motion of our nanomachines, the glass slide was pre-coated with a positively-charged polylysine layer for the immobilization of the negatively charged ICG@CCM-AuNC-PO$_2$-Hb (Fig. 3i). N-STORM (STochastic Optical Reconstruction Microscopy) mode was then used to record the fluorescent changes of our ICG@CCM-AuNC-PO$_2$-Hb before and after laser treatment. As shown by the white arrow in Fig. 3j, few spherical fluorescent signals with size around 30 nm were clearly observed near the nanomachine settled in the bottom right corner after irradiating with 808 nm laser (0.5 W cm$^{-2}$, 5 min). Since fluorescent Alexa 647 was pre-labeled onto the cell membrane, therefore the generated red fluorescent signals with small size were cell membrane-based nanovesicles,

which were triggered by laser shining. Moreover, N-SIM (Structure Illumination Microscopy) mode was also used to follow the dynamic process of nanovesicles secretion. Under this mode, the resolution of the fluorescent signal is up to 115 nm due to the structure lighting and digital image analysis. Although the reso-lution of N-SIM is lower than that of N-STORM, the imaging speed is much faster for better and longer dynamic shooting. ICG@CCM-AuNC and ICG@CCM-AuNC-PO$_2$-Hb were dropped respectively onto the glass slide pre-coated with polylysine and then irradiated with 808 nm laser (0.5 W cm$^{-2}$, 2 min). The obvious generation of nanovesicles was clearly observed in 85 s and the secretion process was completed in 116 s (Fig. 3k, Supplementary movies 1 and 2). As a comparison, no significant change was found for the group of ICG@CCM-AuNC during the laser irradiation (Fig. 3l).

After confirming the laser-triggered nanovesicle secretion of our nanomachines, dark/phototoxicity of ICG@CCM-AuNC-PO$_2$-Hb on 4T1 tumor cells were further investigated. As shown in Fig. 4a, more than 85% of 4T1 cells were dead after treating with ICG@CCM-AuNC-PO$_2$-Hb followed by laser irradiation, probably because of the cell membrane-mediated homotypic targeting and synergetic effect of photodynamic and photothermal therapy. Normal mouse embryo fibroblast cells (NIH3T3 cells) were also used to test the viability of ICG@CCM-AuNC-PO$_2$-Hb (Supplementary Fig. 9). When NIH3T3 cells were treated with the same formulations, their cytotoxicity was generally lower than that in the groups of 4T1 cells, indicating good biocompatibility of our nanodevices. Live (green)/dead (red) staining was then used for the evaluation of in vitro cell toxicity (Fig. 4b). Highest cell apoptosis (red) of ICG@CCM-AuNC-PO$_2$-Hb was clearly observed with the same irra-diation condition when compared with that of other groups. Whereas no significant cells apoptosis was found for all the groups without NIR laser treatments (Supplementary Fig. 10). 3D tumor spheres were further established to verify the antitumor effect of our nanomachines (Supplementary Fig. 11). Similar results were obtained for ICG@CCM-AuNC-PO$_2$-Hb under laser irradiation, indicating the best tumor sup-pression of our design.

Three different cell types including 4T1 cells, melanoma cells (B16-F10 cells), and NIH3T3 cells were used to evaluate the selec-tive uptake of our nanomachines. Fluorescently labelled 4T1 cells (DiO, green) were co-cultured with NIH3T3 cells. After incubating with ICG@CCM-AuNC-PO$_2$-Hb (self-fluorescent ICG, red), a sig-nificant fluorescent signal was clearly observed in the homotypic 4T1 cells, while much less fluorescence was found in NIH3T3 cells (Fig. 4c), indicating homotypic targeting capability of ICG@CCM-AuNC-PO$_2$-Hb. Upon laser irradiation, intracellular red signals in 4T1 cells were significantly increased, probably because of the enhanced uptake of the generated membrane-derived nanove-sicles with smaller sizes by our bubble nanomachines. Flow cyto-metry was further used for the quantification of the intracellular mean fluorescent intensity (MFI). As shown in Fig. 4d and Sup-plementary Fig. 12, for our ICG@CCM-AuNC-PO$_2$-Hb with laser irradiation, MFI of ICG in 4T1 cells was 5 folds higher than that of cells without any treatment, which was another evidence for homotypic targeting and nanovesicle generation of our nano-machines. As non-homotypic tumor cells, B16-F10 cells were also co-cultured with DiO-labeled 4T1 cells to assess the uptake dif-ference between non-homotypic and homotypic cells. Similar

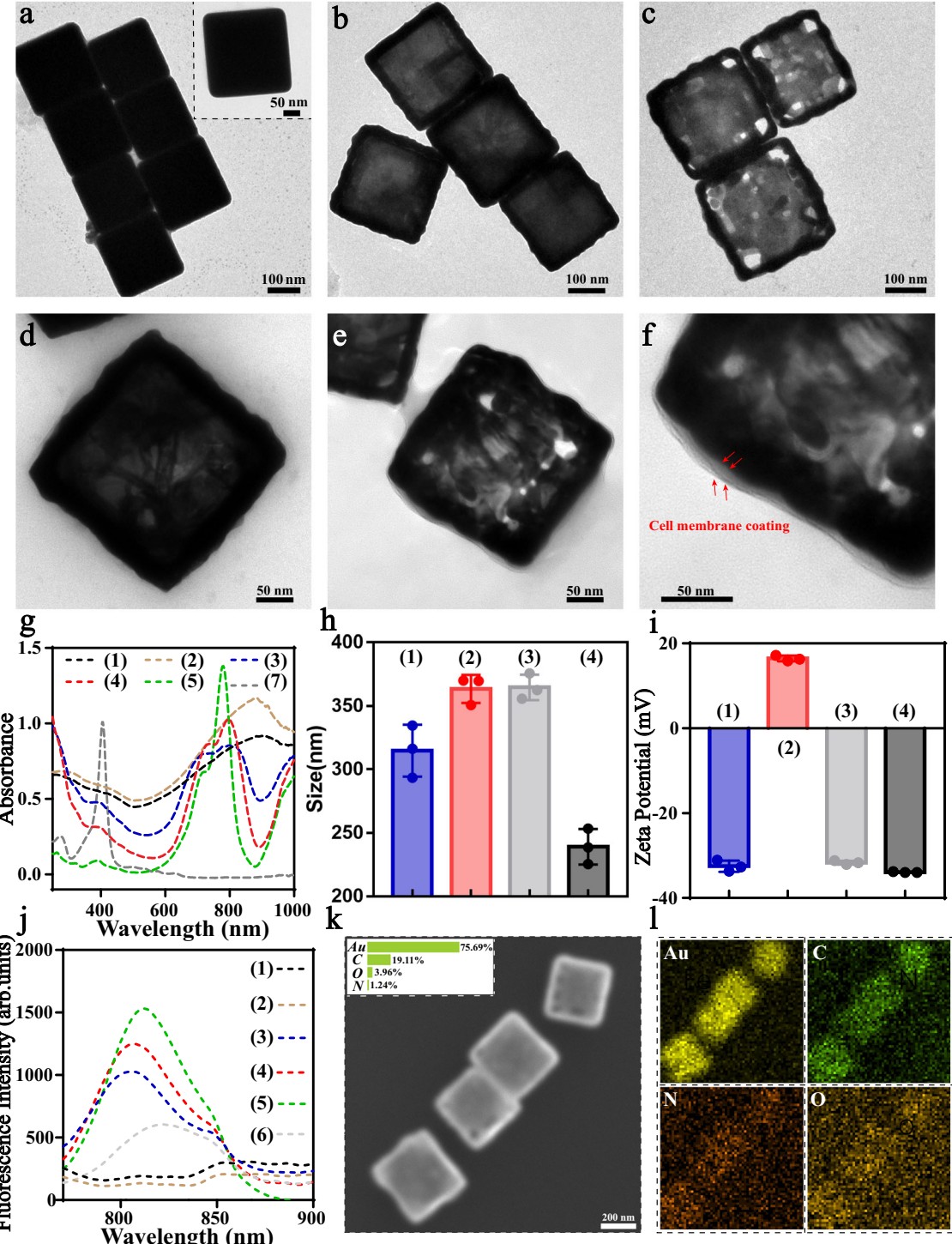

**Fig. 2 | Characterization of ICG@CCM-AuNC-PO2-Hb.** TEM images of **a** AgNC, **b** Ag-AuNC, **c** AuNC, **d** AuNC-PO$_2$-Hb, **e**, **f** ICG@CCM-AuNC-PO$_2$-Hb. **g** UV-vis spectra of ICG@CCM-AuNC-PO$_2$-Hb. **h** Sizes and **i** zeta potentials of ICG@CCM-AuNC-PO$_2$-Hb ($n$ = 3 independent samples). Data are presented as means ± SD. **j** Fluorescent spectra of ICG@CCM-AuNC-PO$_2$-Hb under the excitation of 750 nm. **k** SEM image of ICG@CCM-AuNC-PO$_2$-Hb (inset: atomic percentage of the corresponding nanoparticles). **l** EDX mapping of ICG@CCM-AuNC-PO$_2$-Hb. (1. AuNC, 2. AuNC-PO$_2$-Hb, 3. ICG@CCM-AuNC-PO$_2$-Hb, 4. ICG@CCM, 5. ICG, 6. CCM, 7. Hb). The inside abbreviations [Au] = Gold, [C] = Carbon, [N] = Nitrogen, [O] = Oxygen. Experiments were performed three times (**a**–**f**) or twice (**k**, **l**), with similar results. Source data are provided as a Source Data file.

results (Supplementary Fig. 12) were obtained, which demonstrated that our ICG@CCM-AuNC-PO$_2$-Hb could efficiently distinguish homologous cancer cells from others and further promote the intracellular uptake by nanovesicle secretion. The flow gate strategies of Fig. 4d was shown in Supplementary Fig. 13.

Based on the anchored ICG in our nanomachines, intracellular photodynamic activity of ICG@CCM-AuNC-PO$_2$-Hb under laser irradiation was also evaluated by ROS fluorescent probes namely DCFH-DA. As shown in Supplementary Fig. 14a, higher intracellular ROS fluorescence after ICG@CCM-AuNC-PO$_2$-Hb

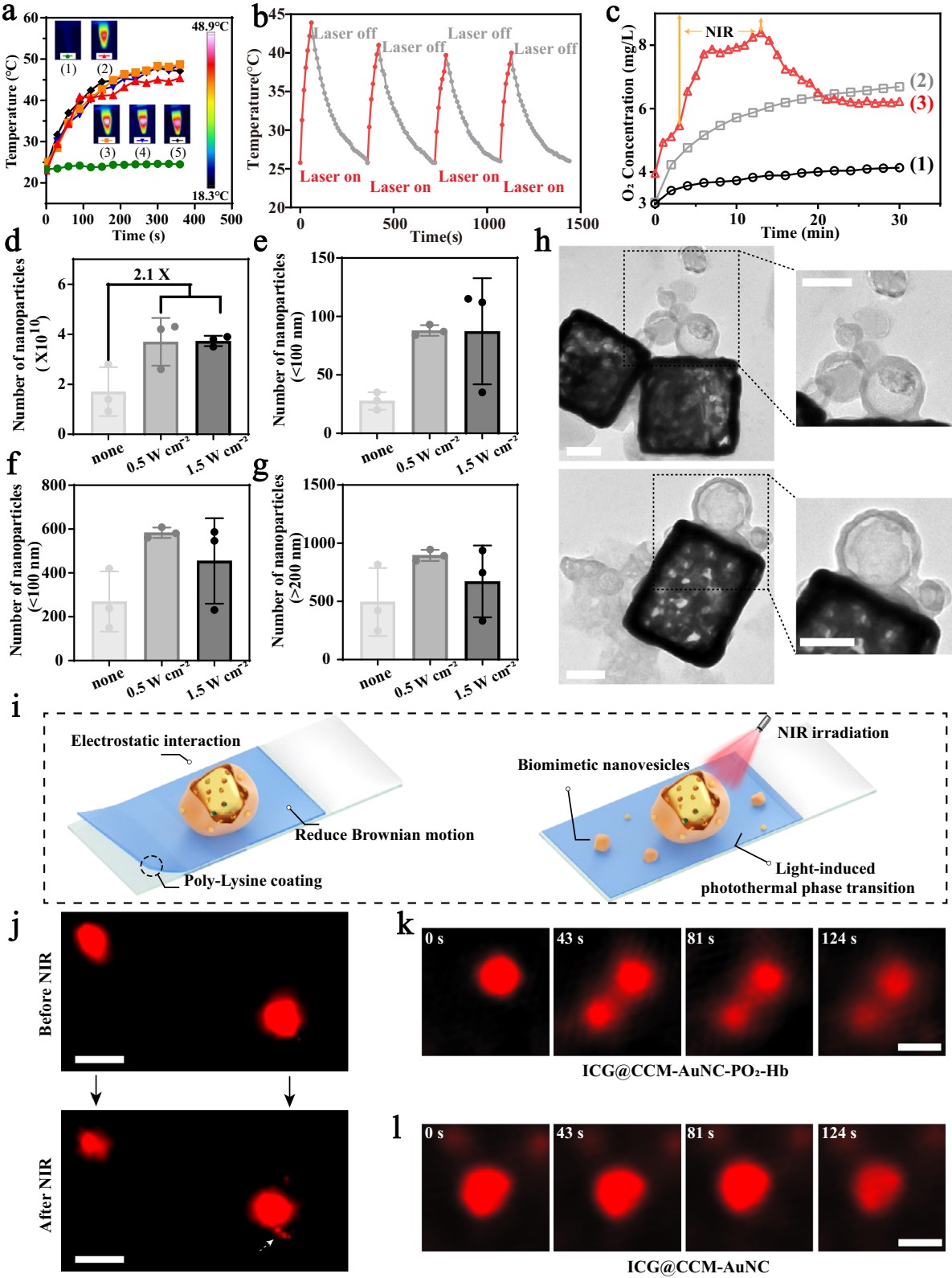

incubation was observed when compared with that of ICG@CCM-AuNC treatment, confirming the promotion of the released exogenous oxygen on photodynamic effect of ICG and the enhanced intracellular uptake of the produced small-scale nanovesicles after NIR irradiation. However, no significant ROS signal was observed for the cells treated with AuNC and AuNC-PO$_2$-Hb because of ICG absence. Similar results were further verified by

the quantitative analysis with flow cytometry (Supplementary Fig. 14b). In particular, the fluorescent intensity of the group treated with ICG@CCM-AuNC-PO$_2$-Hb was 1.6 folds higher than that of ICG@CCM-AuNC group without oxygen release and nanovesicle secretion (Supplementary Fig. 14c). The flow gate strategies of Supplementary Fig. 14b was shown in Supplementary Fig. 15.

**Fig. 3 | Photothermal properties and photothermal-induced biomimetic nanovesicle secretion of the bubble machine. a** Thermal images and temperature increase curve of (1). PBS, (2). AuNC, (3). AuNC-PO$_2$-Hb, (4). CCM-AuNC-PO$_2$-Hb and (5). ICG@CCM-AuNC-PO$_2$-Hb under 808 nm laser irradiation (1 W cm$^{-2}$). The embedded figures were represented the final temperature infrared thermal images of different preparations. **b** Photostability of ICG@CCM-AuNC-PO$_2$-Hb. **c** O$_2$ concentration after addition of ICG@CCM-AuNC or ICG@CCM-AuNC-PO$_2$-Hb into the deoxygenated water with or without NIR irradiation (808 nm, 1 W cm$^{-2}$). (1) ICG@CCM-AuNC, (2) ICG@CCM-AuNC-PO$_2$-Hb, (3) ICG@CCM-AuNC-PO$_2$-Hb + NIR. **d–g** Nanoparticle number of ICG@CCM-AuNC-PO$_2$-Hb group in different size

ranges after laser irradiation (none, 0.5 W cm$^{-2}$, 1.5 W cm$^{-2}$) obtained from Zeta-Viewer ($n = 3$ independent samples). Data are presented as means ± SD. **h** TEM images of ICG@CCM-AuNC-PO$_2$-Hb after NIR irradiation (808 nm, 0.5 W cm$^{-2}$) for 1.5 min, scale bar = 100 nm. **i** Illustration of the set-up for N-STORM and N-SIM measurements. **j** N-STORM images of ICG@CCM-AuNC-PO$_2$-Hb obtained by super-resolution fluorescence microscopy before and after NIR irradiation (0.5 W cm$^{-2}$, 5 min), scale bar = 200 nm. N-SIM images of **k** ICG@CCM-AuNC-PO$_2$-Hb and **l** ICG@CCM-AuNC during laser irradiation (808 nm, 0.5 W cm$^{-2}$), scale bar = 1 μm. Experiments were performed three times (**h**, **j–l**), with similar results. Source data are provided as a Source Data file.

Based on the laser-triggered secretion of nanovesicles with smaller size, we assumed that this special capability of our ICG@CCM-AuNC-PO$_2$-Hb was possible to promote tumor penetration. Therefore, 3D multicellular tumor spheroids based on 4T1 and NIH3T3 cells were established for the evaluation of tumor penetrability of our nanomachines. As shown in Fig. 4e, ICG@CCM-AuNC-PO$_2$-Hb incubation followed by laser treatment showed the best penetration depth in 3D tumor spheroid, which was attributed to the generation of nanovesicles with smaller size. For the group of ICG@CCM combined with laser treatment, tumor penetration was also enhanced due to the smaller particle sizes, but most of the nanoparticles just remained on the surface of the tumor sphere, especially at 100 μm depth. Furthermore, the penetrating percentage was calculated based on the fluorescent intensity ratio between the inner and edge region (Supplementary Fig. 16). The produced nanovesicles based on 4T1 tumor cell membrane were capable of permeating nearly 93% of a tumor spheroid, even in 100 μm depth from the surface to the equatorial plane. Besides, the penetration percentage of ICG@CCM with NIR was less than 50%, possibly because of no in situ photothermal promotion of penetration. Whereas for either ICG@CCM-AuNC-PO$_2$-Hb without laser irradiation or ICG@CCM-AuNC with laser treatment, ICG signal was only observed on the surface of tumor spheroid in the depth of 100 μm, indicating superb penetrability of our bubble nanomachines.

Encouraged by the excellent in vitro homotypic tumor-targeting capability and tumor penetrability of our bubble nanomachines with laser-triggered nanovesicle secretion, in vivo evaluation of ICG@CCM-AuNC-PO$_2$-Hb was further investigated. Free ICG, ICG-AuNC-PO$_2$-Hb, and ICG@CCM-AuNC-PO$_2$-Hb with the same ICG concentration of 45 μg mL$^{-1}$ were firstly administrated intravenously and in vivo imaging of tumor-bearing mice (BALB/c) was then recorded at different time intervals. For our ICG@CCM-AuNC-PO$_2$-Hb, a strong fluorescent signal was observed in major organs including liver and spleen in the beginning and it was transferred gradually into tumor region after injecting for 8 h because of homotypic tumor targeting (Fig. 5a). Whereas, for the mice treated with free ICG and ICG-AuNC-PO$_2$-Hb, the negligible signal was found in the tumor, which was further confirmed by ex vivo imaging of the collected organs and tumors (Fig. 5b). It has been found that the collected tumor from ICG@CCM-AuNC-PO$_2$-Hb group exhibited much higher fluorescent intensity than that of ICG-AuNC-PO$_2$-Hb without 4T1 cell membrane cloaking. In vivo distribution map of ICG@CCM-AuCN-PO$_2$-Hb after laser irradiation was also demonstrated. After 24 h of injection, all the mice were recorded by in vivo fluorescence imaging. The mice were irradiated with laser for 1.5 min immediately after imaging, followed by another fluorescence imaging at 26 h. As shown in the Supplementary Fig. 17, the increase in fluorescence intensity was clearly observed in the tumor after laser irradiation. It is another piece of evidence to prove that photothermal effect can induce the formation of smaller-sized nanovesicles to promote effective tumor penetration. In vivo photothermal imaging of tumor-bearing mice was further conducted. Upon post-administration of ICG@CCM-AuNC-PO$_2$-Hb for 24 h, the tumor region was then irradiated with NIR laser (808 nm, 1.0 W cm$^{-2}$) for 3 min. As shown in Supplementary Fig. 18, the local temperature of tumor site was rapidly increased to 62 °C within 3 min after irradiation. However, a slight

temperature increase was observed for AuNC-PO$_2$-Hb and CCM-AuNC-PO$_2$-Hb, probably because of either NIR absorbance by surrounding tissues or the absence of ICG anchoring.

In vivo antitumor efficacy of our bubble nanomachines was further investigated with 4T1 xenograft mammary tumor bearing BALB/c mice (female, 6 weeks). The mice were divided into 6 groups ($n = 5$) including (1) PBS, (2) ICG@CCM-AuNC + NIR, (3) AuNC-PO$_2$-Hb + NIR, (4) CCM-AuNC-PO$_2$-Hb + NIR, (5) ICG@CCM-AuNC-PO$_2$-Hb, (6) ICG@CCM-AuNC-PO$_2$-Hb + NIR, and the mice were administrated on day 3 and day 12, respectively (Fig. 5c). The tumor regions were then irradiated for 1.5 min with NIR laser (808 nm, 0.5 W cm$^{-2}$) after administration for 24 h, followed by laser irradiation (1 W cm$^{-2}$) for 3 min after another 1 h. The purpose of the first 1.5 min laser treatment was to trigger the generation of small-scale nanovesicles for enhanced tumor penetration, while the subsequent 3 min laser irradiation was utilized for photothermal and photodynamic therapy. The tumor sizes were recorded every two days for 24 days, after which the tumors were collected and weighted. As shown in Fig. 5d–f, and Supplementary Fig. 19, the combination of ICG@CCM-AuNC-PO$_2$-Hb with laser irradiation displayed the best antitumor efficacy and the tumor volume was suppressed considerably by 97% when compared with that of PBS group. Even the tumors of 60% mice administrated with our bubble nanomachines disappeared after treatment. This impressive antitumor efficacy was attributed to the enhanced tumor penetration of the produced nanovesicles with smaller size from ICG@CCM-AuNC-PO$_2$-Hb under laser irradiation, and the synergistic effect of photothermal and photodynamic therapy of both generated nanovesicles and ICG@CCM-AuNC-PO$_2$-Hb itself. However, the group of ICG@CCM-AuNC-PO$_2$-Hb without laser irradiation hardly showed a tumor inhibitory effect because of absent photothermal or photodynamic therapy. In order to fully demonstrate the efficacy of our nanomachine, orthotopic tumor models were further established. Similar results were obtained and the mice treated with ICG@CCM-AuNC-PO$_2$-Hb combined with NIR showed the best antitumor effect (Supplementary Fig. 20a). Due to the deep photothermal and photodynamic therapy generated by our bubble nanomachine, most of the orthotopic breast tumors were completely disappeared in volume (Supplementary Fig. 20b, c). To further investigate the photodynamic therapy-induced effect of our synergistic strategy, hypoxia-inducible factor (HIF−1α) staining assay was further performed to evaluate the in vivo hypoxia level in tumor. As demonstrated in Supplementary Fig. 21, the group treated with PBS displayed stronger green immunofluorescence signal because of the hypoxic condition under tumor microenvironment. For the mice treated with ICG@CCM-AuNC-PO$_2$-Hb with O$_2$ supply, much weaker fluorescence signal was observed, which meant a low-level expression of HIF-1α was induced in tumor tissue.

The body weights of the mice during the treatment were also monitored and no significant changes were observed, indicating no acute toxicity of our bubble nanomachines (Supplementary Fig. 22). Moreover, H&E staining of the collected tumors and main organs including the heart, liver, spleen, lung, and kidney was then conducted. As shown in Fig. 5g, nearly all the tumor cells underwent apoptosis after treatment with our ICG@CCM-AuNC-PO$_2$-Hb capable of nanovesicle secretion triggered by laser illumination.

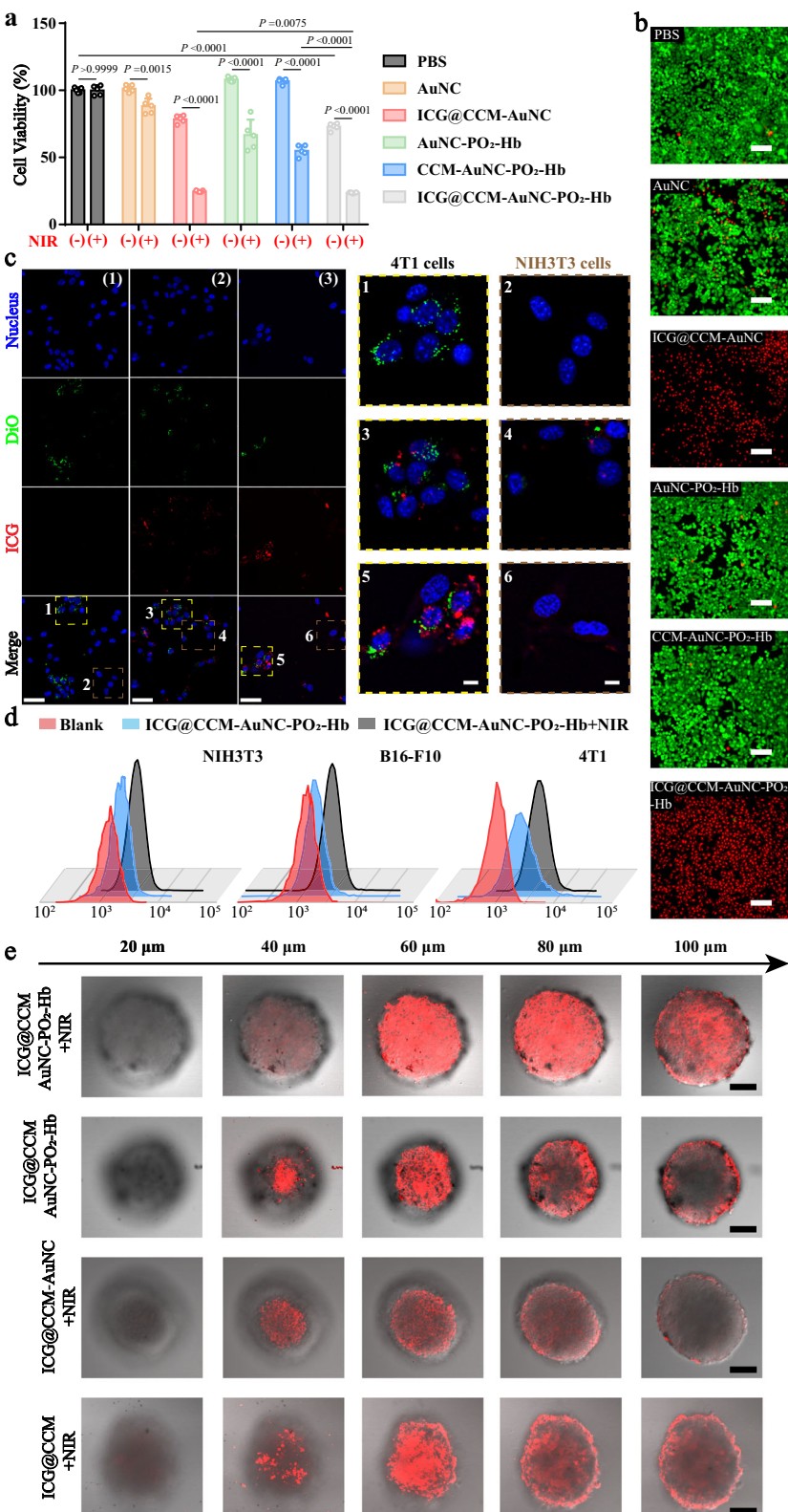

For H&E staining of the main organs, no obvious morphological changes and inflammatory infiltrates were observed, indicating long-term biosafety of our bubble nanomachines (Supplementary Fig. 23). Blood routine and serum biochemical parameters of all the groups were determined as well (Fig. 5h-j, Supplementary Fig. 24). As expected, there was no significant changes for all the parameters, demonstrating the superior biocompatibility of our design from another aspect. Furthermore, blood compatibility of ICG@CCM-AuNC-PO$_2$-Hb with different concentrations (1000, 500, 200, 100, and 20 µg mL$^{-1}$) was also examined by using hemolysis test. As shown in Supplementary Fig. 25, less than 4% hemolysis rate was observed even the concentration of our bubble nanomachines increased to 1000 µg mL$^{-1}$, thus indicating excellent cytocompatibility and blood compatibility.

**Fig. 4 | Dark/photo-toxicity and homologous targeting in cells and penetration of tumor spheres. a** Dark and phototoxicity of 4T1 cells after different treatments. [NIR (-)] represented without 808 nm laser irradiation; [NIR (+)] represented with 808 nm laser irradiation. Data are given as means ± SD ($n$ = 5 biologically independent samples). Significant differences were evaluated by two-tailed unpaired $t$-test. **b** Fluorescence images of live/dead staining of 4T1 cells. The live cells were stained with AM (green) and the dead cells were stained with PI (red). Scale bar =100 μm **c** Homotypic targeting of ICG@CCM-AuNC-PO$_2$-Hb on co-cultured 4T1/NIH3T3 cells (blue = nucleus, green = 4T1 cells membrane, red = ICG@CCM-AuNC-PO$_2$-Hb). (1) blank, (2) ICG@CCM-AuNC-PO$_2$-Hb and (3) ICG@CCM-AuNC-PO$_2$-Hb + NIR. Scale bar = 50 μm (left), scale bar = 10 μm (right). The images on the right represent enlargements of the numbers corresponding to the embedded boxes in the image on the left, respectively. **d** Flow cytometric images of 4T1, NIH3T3 and B16-F10 cells after incubation with ICG@CCM-AuNC-PO$_2$-Hb, followed by 808 nm laser irradiation for 1.5 min (0.5 W cm$^{-2}$). **e** In vitro penetration of ICG@CCM-AuNC-PO$_2$-Hb after laser irradiation. Z-stack images using CLSM with thickness of 20 μm were obtained from the top to the equatorial plane of 3D 4T1/ NIH3T3 tumor spheroids. Scale bar = 100 μm. Experiments were performed three times (**b**–**e**), with similar results. Source data are provided as a Source Data file.

In summary, we demonstrated the successful design and fabrication of biomimetic bubble nanomachines capable of generating tumor cell-membrane-derived nanovesicles with smaller size for enhanced tumor penetration to achieve synergistic photothermal/photodynamic therapy. Phase transitable PFO and Hb with oxygen pre-saturation were firstly loaded in the cavity of porous AuNCs and 4T1 tumor cell membrane-anchored with ICG was then cloaked, demonstrating improved tumor accumulation based on homologous targeting effect with reduced systemic toxicity. Upon NIR laser irradiation, AuNCs in situ generated heat due to the surface plasmon resonance effect, thus inducing phase transition of the loaded PFO. Gasified PFO was able to cross the pores of AuNCs and extrude the coated cell membrane to form nanovesicles with smaller size, resulting in enhanced tumor penetration and synergistic photothermal and photodynamic therapy. Learn from daily life, our work thus presents a smart strategy to in-situ generate cell membrane-derived nanovesicles in the local tumor region. The as-developed laser-triggered bubble nanomachines hold considerable promising as potential platform for the treatment of solid tumors.

## Methods

### Cell lines and animals

NIH3T3 (Mouse embryonic fibroblast) cells were kindly provided by Dr. Weichang Huang from The First Clinical Medical College of Southern Medical University and B16-F10 (Mouse melanoma) cells were obtained from the ATCC (CRL-6475), which were cultured in high glucose DMEM complete medium (Procell, PM150210B). 4T1 (Mouse mammary tumor) cells (Lot: XRBETC2TZM) were purchased from Procell Life Science&Technology Co., Ltd and cultured in RPMI-1640 complete medium (Procell, CM-0007). The cells were cultured in a 5% CO$_2$ humidified incubator at 37 ˚C. Female BALB/c mice (4-6 weeks old, 18–22 g) and BALB/c nude mice (5-6 weeks old, 17–20 g) were obtained from the Laboratory Animal Center of Southern Medical University. The mice had access to food and water ad libitum and were hosted in ambient temperature (22–24 °C), humidity at 30–70%, under 12 h dark/light cycles. All the animal procedures were carried out under the guideline approved by the Institutional Animal Care and Use Committee (IACUC) of Southern Medical University (permit number: SMUL2022180).

### Materials

Sodium hydrosulfide (NaHS), chloroauric acid (HAuCl$_4$), hemoglobin (Hb), hydrochloric acid (HCl) and acetone were bought from Innochem. Ethylene glycol (EG), silver nitrate (AgNO$_3$) and 3,3'-Dioctadecyloxacarbocyanine perchlorate (DiO) were obtained from Aladdin. Poly(viny pyrrolidone) (PVP, $M_w \approx 55,000$) was purchased from Sigma-Aldrich (St. Louis). Perfluorohexane (PFO) was bought from Energy Chemical. Silver trifluoroacetate (CF$_3$COOAg) was obtained from Adamas. Indocyanine green (ICG) was purchased from ARK Pharm, Inc. Hoechst 33342, Cytosol Protein Extraction Kit and phenylmethanesulfonyl fluoride (PMSF) were purchased from Beyotime Biotechnology. All reagents for cell culture were bought from Gibco. The purified deionized water was prepared by the Milli-Q plus system (Millipore, USA).

### Instruments

The structures of the formed nanoparticles were analyzed with a FEI Tecnai G2 F30 Transmission Electron Microscope (TEM) with an acceleration voltage of 300 kV. Scanning electron microscope (SEM) images were recorded on a Phenom emission scanning electron microscope. Energy-dispersive X-ray spectroscopy (EDX) Element analysis was carried out with an EDX analyzer (mounted on the Phenom ProX) with an accelerating voltage of 15 kV. Zeta potential and size measurements were performed on Zetasizer Nano ZSE (Malvern, UK) with the following settings: temperature 25 °C, He-Ne laser wavelength 633 nm. The particle number were investigated with Nanoparticle Tracking Analysis (NTA) based on Zetaview (PMX, Germany). The UV-vis absorption was measured with a UV-2600 spectrophotometer (SHIMADZU, Japan). Fluorescence spectra were analyzed with a RF-6000 fluorescence spectrophotometer (SHIMADZU, Japan). Cell morphologies were captured on a Ti2-A Inverted Fluorescence Microscope (Nikon, Japan) and LSM 880 with Airyscan confocal microscope (Carl Zeiss, Germany). The mean fluorescence intensity of cells was analyzed with cytoFLEX (BECKMEN, USA). The thermographic images were acquired by an infrared thermal camera (FLIR, USA). Photothermal therapy on cells or mice were carried out with 808 nm laser (Stone, China). In vivo imaging of mice or collected organs was performed by In-Vivo FX PRO (BRUKER, Germany). All the parameters of blood biochemistry were analyzed by Pointcare M3 (MNCHIP, China).

### Synthesis of Ag nanocubes

Seed-mediated growth method was used for the synthesis of Ag nanocubes (AgNC)[31]. EG (5 mL) was firstly heated under magnetic stirring in an oil bath preset to 150 °C. NaSH (10.1 μg, 0.0001802 mmol in EG) was added after 10 min and a 3 mM HCl solution (0.5 mL, 2.5 μL HCl in 10.3 mL EG solution) was then added 2 min later, followed by the addition of 1.25 mL PVP solution (25 mg, 0.000455 mmol in EG). After another 2 min, 0.4 mL CF$_3$COOAg (24.92 mg, 0.1128 mmol in EG) was then added and the color of the resulting mixture quickly became slightly yellow in 1 min. After another 40 min, the reaction was quenched by placing the flask in an ice-water bath. The samples were collected by centrifugation (6080 × $g$, 15 min) and then washed with acetone (to remove EG)/DI water (to remove excess PVP) for four times. Ag seeds (38% yield) were then stored in EG and the particle concentration was determined by NTA.

1.25 mL of EG was heated to 150 °C and 0.3 mL of a PVP solution in EG (6 mg, 0.0001092 mmol) was added after 1.5 min. After another 1.5 min, 50 μL of the as-prepared single-crystal Ag seeds (1.1 × 10$^{12}$ particles/mL) in EG solution was then added, followed by the addition of 200 μL of AgNO$_3$ solution in EG (9.58 mg, 0.0564 mmol). After 1 h, the growth was terminated by quenching with an ice-water bath. The resulting mixtures were centrifuged (860 × $g$), and washed with acetone and DI water, respectively. The obtained AgNC (40.9% yield) were suspended in DI water (2 mL) for further usage.

### Synthesis of Au nanocages

200 μL of AgNC solution was dispersed into 10 mL of PVP aqueous solution (10 mg, 0.0001818 mmol). Then the resulting mixture was heated to 95 °C, followed by the dropwise addition of HAuCl$_4$

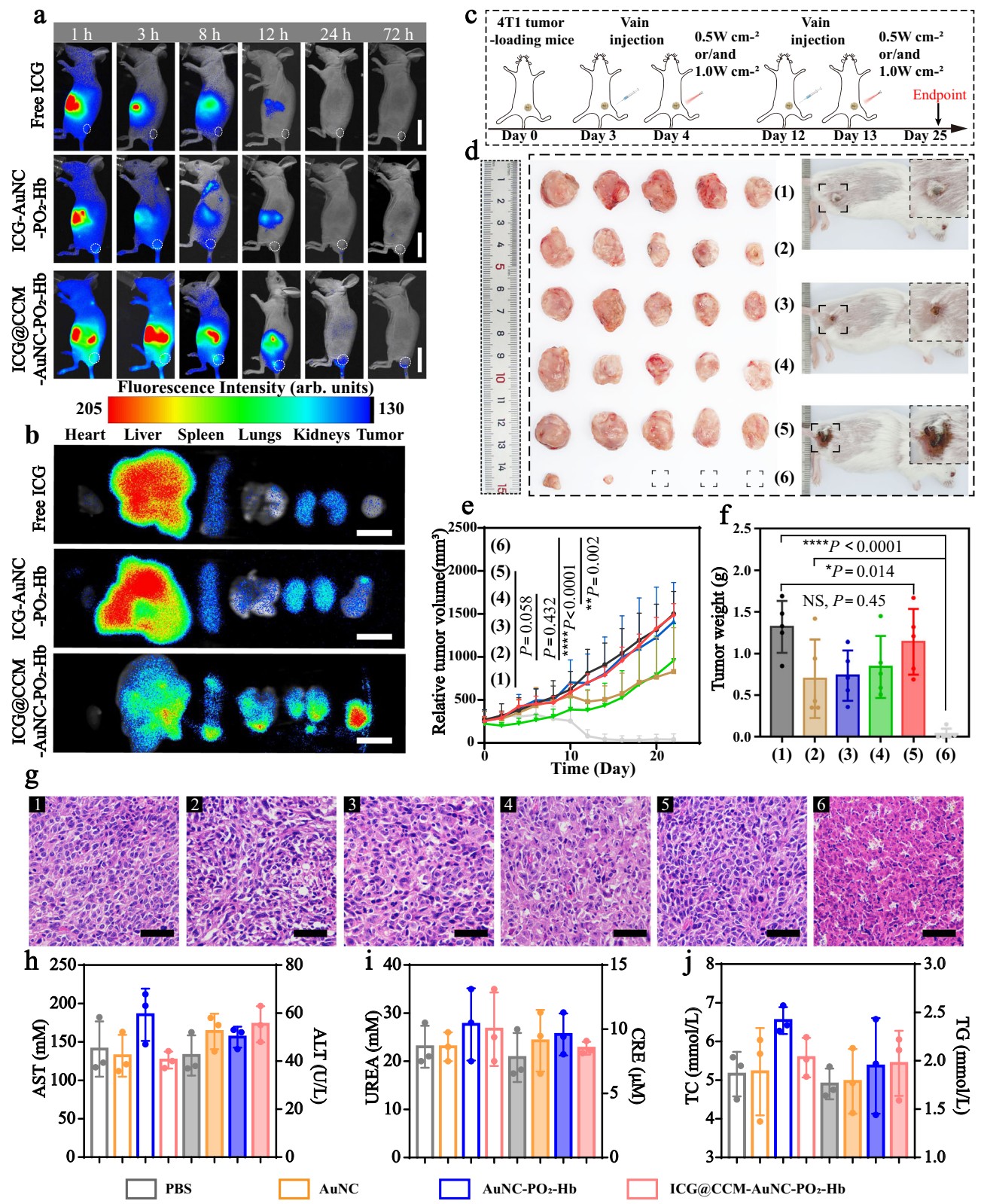

solution (0.01269 µmol mL$^{-1}$) at the rate of 0.75 mL min$^{-1}$. The reaction was continued until the LSPR spectrum of the resulting mixture shifted to the desired wavelength of 880 nm. The mixture was then purified by centrifugation at 3420 × $g$ for 10 min. The obtained nanoparticles were dispersed in 2 mL of ultrapure water and frozen overnight. The frozen samples were transferred to a vacuum freeze-drying oven (set at −40 °C) and dried for 24 h.

Finally, the obtained Au Nanocages (AuNC, 79% yield) were stored as powder after lyophilization.

## PFO loading and oxygen saturation

AuNC (1.0 mg) powder was added into a 50 mL bottle and the resulting bottle was evacuated with a vacuum pump[37]. Then 200 µL of PFO was injected into the bottle and sonicated for 2 min under ice-water bath.

**Fig. 5 | In vivo evaluation of ICG@CCM-AuNC-PO2-Hb with nanovesicle secretion. a** In vivo distribution of free ICG, ICG-AuNC-PO$_2$-Hb and ICG@CCM-AuNC-PO$_2$-Hb in tumor-bearing mice at different time points. Scale bars =2 cm. **b** Ex vivo imaging of the collected organs and tumors from the mice after injection for 24 h. Scale bars =1 cm. **c** Therapeutic regimen of 4T1 tumor-bearing mice with ICG@CCM-AuNC-PO$_2$-Hb. **d** Collected tumors (left) and representative images of mice (right) from group (6). Embedded boxes (right) are images representing their (small boxes on left) enlargements, respectively. **e** Tumor volumes of 4T1 tumor-bearing mice during the treatments. **f** Tumor weights of the mice after administrating our bubble nanomachines. **g** H&E staining of the collected tumors (scale bars = 50 μm). **h–j** Alanine aminotransferase (ALT), aspartate aminotransferase (AST), blood urea nitrogen (UREA), creatinine (CRE), total cholesterol (TC) and triglyceride (TG) of 4T1 tumor-bearing mice after treatments. (1) PBS; (2) ICG@CCM-AuNC + NIR; (3) AuNC-PO$_2$-Hb + NIR; (4) CCM-AuNC-PO$_2$-Hb + NIR; (5) ICG@CCM-AuNC-PO$_2$-Hb; (6) ICG@CCM-AuNC-PO$_2$-Hb + NIR. For 5 **e**, **f**, $n$ = 5 biologically independent animals. For 5 **h–j**, $n$ = 3 biologically independent samples. Data are presented as means ± SD. Significant differences were evaluated by two-tailed unpaired $t$-test. Experiments were performed three times (**g**), with similar results. Source data are provided as a Source Data file.

After centrifugation (860 × $g$, 5 min) to remove the excess PFO, 1 mL of PBS was injected and sonicated for another 5 min under ice-water bath. Hb solution (0.5 mg mL$^{-1}$) was then added into PFO loaded AuNC (1 mL) and the resulting mixture was stirred for 6 h (20 × $g$). After centrifugation, the obtained AuNC-PFO-Hb was dispersed in 1 mL PBS and stored in an aseptic oxygen chamber for 10 min to achieve oxygen saturation. Oxygen-saturated, Hb and PFO-loaded AuNC was referred to as AuNC-PO$_2$-Hb.

### ICG anchoring and cell Membrane coating
To obtain 4T1 cancer cell membrane vesicles (CCMs), 4T1 cells were centrifuged for 5 min (860 × $g$, 4 °C) and then resuspended in hypotonic lysing buffer (containing membrane protein extraction reagent and PMSF). After incubating with an ice bath for 12 min, the resulting cells were transferred into Dounce homogenizer for mechanical disruption (40 times). Afterwards, the mixtures were centrifuged (700 × $g$) at 4 °C for 12 min and the resulting supernatant was centrifuged again with (14,000 x $g$) for 40 min. The final 4T1 cell membrane fragments were collected and resuspended in PBS to prepare membrane stock solution with 1 mg mL$^{-1}$. For anchoring ICG in membrane, 1 mL above-montioned stock membrane solution was incubated with ICG solution (1 mL, 0.5 mg mL$^{-1}$ dissolved in ultrapure water) with gently stirring (4 °C, 300 rpm) for 3 h at dark. After centrifuging (16060 × $g$, 30 min), ICG anchored CCMs (ICG@CCM segments) was collected.

Before use, ICG anchored cell membrane fragments (1 mg) were redispersed in 1 mL PBS (0.1 M, pH = 7.4) and sonicated for 10 s, and then extruded through 800 nm and 400 nm polycarbonate membranes (LiposoFast, Avestin, Canada) at least 7 times respectively to obtain ICG@CCM vesicles. AuNC-PO$_2$-Hb (0.5 mL, 1 mg mL$^{-1}$) were then mixed with the resulting ICG@CCM (1 mL, 1 mg mL$^{-1}$) solution to co-extrude via 400 nm polycarbonate membranes at least for 5 circles. Then, the mixture was centrifuged (5000 × $g$) at 4 °C for 20 min to achieve final formulation (ICG@CCM-AuNC-PO$_2$-Hb).

### Characterization of ICG@CCM-AuNC-PO$_2$-Hb
The content of Au was quantified by measuring Au$^{3+}$ concentration using Inductively Coupled Plasma-Mass Spectrometry (ICP-MS) (Thermo Fisher Scientific). 100 μL of the sample solution (ICG@CCM-AuNC-PO$_2$-Hb, 300 μg mL$^{-1}$) was added to 4.5 mL of nitric acid and placed in automatic microwave running on the program digestion (120 °C hold for 5 min, 150 °C hold for 5 min, 180 °C hold for 12 min). Then, ultrapure water was added to volume up to 10 mL, from which 0.5 mL of the sample was diluted to 5 mL and used for ICP-MS testing.

The standard curve of free Hb was used to quantify the amount of Hb loading at a UV-Vis wavelength of 405 nm. The standard curve of free ICG was used to quantify the amount of ICG anchored into the cell membrane.

To detect the stability of Hb, 1 mL AuNC-PO$_2$-Hb (1 mg mL$^{-1}$) was shaken in a water bath at 37 °C for 24 h, then centrifuged (3420 × $g$) for 8 min and the supernatant was collected as the experimental group. For the control group, 50 μg of Hb was dissolved in 1 mL of PBS. Circular dichroism (CD) of the samples were measured with Chirascan spectropolarimeter (Applied Photophysics) in a 0.1-cm cuvette (200–260 nm). The secondary structure was computed by DichroWeb online analysis.

To measure the oxygen loading capacity of ICG@CCM-AuNC-PO$_2$-Hb, 5 mL of deoxygenated water was added in a 50 mL centrifuge tube that was closed by a rubber plug. Oxygen electrode probe was inserted to measure the oxygen concentration of the solution in real time. Then, 0.2 mL of oxygen-saturated ICG@CCM-AuNC-PO$_2$-Hb with different concentrations (0, 0.25, 0.5, 1.0 mg mL$^{-1}$) were injected into this closed system. The oxygen concentration was recorded when the oxygen concentration reached equilibrium.

### Proteins analysis of biomimetic nanomachine and the released nanovesicles
SDS-PAGE analysis was performed to estimate the retained proteins of 4T1 cell membrane, ICG@CCM-AuNC-PO$_2$-Hb, Hb and the released nanovesicles. The materials were prepared in SDS sample buffer, and the samples were then run on gradient gel (10%, PAGE Gel Fast Preparation Kit, Epizyme) using BIO-RAD Electrophoresis System. The samples were run at 90 V for 20 min and then at 120 V for 1 h to separate proteins. Next, the resulting polyacrylamide gel was stained by Coomassie brilliant blue for 15 min.

### Measurement of the released O$_2$
Oxygen concentration in aqueous solutions was measured using a portable dissolved oxygen meter (Rex, JPBJ-608, China). 5 mL of deoxygenated water was firstly added into a 50 mL of centrifuge tube filled with nitrogen and 2 mL of peanut oil was then added gently for oxygen isolation. Oxygen electrode probe was then inserted into the tube for the real-time measurement of oxygen concentration. 200 μL of ICG@CCM-AuNC-PO$_2$-Hb or ICG@CCM-AuNC solution (with same Au concentration of 1 mg mL$^{-1}$) was injected into the tube, respectively. Oxygen concentration was recorded for 30 min. For the NIR triggered O$_2$ release, the sample was irradiated by a NIR laser (808 nm, 1.0 W cm$^{-2}$) for 10 min.

### In vitro photothermal performance
PBS, AuNC, AuNC-PO$_2$-Hb, CCM-AuNC-PO$_2$-Hb and ICG@CCM-AuNC-PO$_2$-Hb with same Au concentration (100 μg mL$^{-1}$) were added into EP tube, respectively. All the samples were treated with NIR laser (808 nm, 1 W cm$^{-2}$) for 6 min, following with infrared imaging for each 10 s. To study the photostability of the resulting materials, ICG@CCM-AuNC-PO$_2$-Hb (200 μg mL$^{-1}$) were irradiated with NIR laser (808 nm, 1 W cm$^{-2}$) for 4 cycles under infrared imaging recording.

### In vitro photodynamic properties
In vitro photodynamic properties of ICG@CCM-AuNC-PO$_2$-Hb were evaluated by a typical SOSG assay. Briefly, 1 mL solution (PBS, AuNC, AuNC-PO$_2$-Hb, ICG@CCM-AuNC, CCM-AuNC-PO$_2$-Hb or ICG@CCM-AuNC-PO$_2$-Hb with same Au concentrations of 100 μg mL$^{-1}$) containing 5 μM SOSG was added into the cuvette, and irradiated with NIR laser (808 nm) at the power density of 1.0 W cm$^{-2}$ for 2 min. The fluorescence intensity was recorded by RF-6000 fluorescence spectrophotometer at the peak 525 nm of emission spectrum.

## Calculation of the photothermal conversion efficiency

The photothermal conversion efficiency of ICG@CCM-AuNC-PO$_2$-Hb was determined according to linear time data versus $-\ln\theta$ obtained from the cooling period[42]. ICG@CCM-AuNC-PO$_2$-Hb aqueous dispersion (200 µg mL$^{-1}$, 0.2 mL) were exposed to laser irradiation (808 nm, 1 W cm$^{-2}$) for 390 s and then naturally cooled for 13 min. Detailed calculation was given as followed:

$$\sum_i m_i C_{p,i} \frac{dT}{dt} = Q_{NPs} + Q_s - Q_{loss} \tag{1}$$

where $m$ and $Cp$ are the mass and heat capacity of solvent (water), respectively. T is the solution temperature. $Q_{NPs}$ is the photothermal energy input by ICG@CCM-AuNC-PO$_2$-Hb:

$$Q_{NPs} = I\left(1 - 10^{-A_\lambda}\right)\eta \tag{2}$$

where $I$ is the laser power, $A_\lambda$ is the absorbance of ICG@CCM-AuNC-PO$_2$-Hb at the wavelength of 808 nm, and $\eta$ is the conversion efficiency from the absorbed light energy to thermal energy.

$Q_s$ is the heat associated with the light absorbance of the solvent, which is measured independently to be 1.071 mW using pure water without ICG@CCM-AuNC-PO$_2$-Hb.

$Q_{loss}$ is thermal energy lost to the surroundings:

$$Q_{loss} = hA\Delta T \tag{3}$$

where $h$ is the heat transfer coefficient, $A$ is the surface area of the container, and $\Delta T$ is the temperature change, which is defined as $T$-$T_{surr}$ ($T$ and $T_{surr}$ are the solution temperature and ambient temperature of the surroundings, respectively).

At the maximum steady-state temperature, the heat input is equal to the heat output, that is:

$$Q_{NPs} + Q_s = Q_{loss} = hA\Delta T_{max} \tag{4}$$

where $\Delta T_{max}$ is the temperature change at the maximum steady-state temperature. According to the Eq. 2 and Eq. 4, the photothermal conversion efficiency ($\eta$) can be determined:

$$\eta = \frac{hA\Delta T_{max} - Q_s}{I\left(1 - 10^{-A_\lambda}\right)} \tag{5}$$

In this equation, only $hA$ is unknown for calculation. In order to get the $hA$, we herein introduce $\theta$, which is defined as the ratio of $\Delta T$ to $\Delta T_{max}$:

$$\theta = \frac{\Delta T}{\Delta T_{max}} \tag{6}$$

Substituting Eq. 6 into Eq. 1 and rearranging Eq. 1:

$$\frac{d\theta}{dt} = \frac{hA}{\sum_i m_i C_{p,i}}\left[\frac{Q_{NPs} + Q_s}{hA\Delta T_{max}} - \theta\right] \tag{7}$$

When the laser was shut off, the $Q_{NPs} + Q_s = 0$, Eq. 7 changed to:

$$dt = -\frac{\sum_i m_i C_{p,i}}{hA}\frac{d\theta}{\theta} \tag{8}$$

Integrating Eq. 8 gives the expression:

$$t = -\frac{\sum_i m_i C_{p,i}}{hA}\theta \tag{9}$$

Thus, $hA$ can be determined by applying the linear time data from the cooling period vs $-\ln\theta$ (Supplementary Figure. 15). Substituting $hA$ value into Eq. 5, the photothermal conversion efficiency ($\eta$) of ICG@CCM-AuNC-PO$_2$-Hb can be calculated (to be ~ 26.81 ± 1.3%).

## Particle number evaluation of the generated nanovesicles triggered by NIR irradiation

ICG@CCM-AuNC-PO$_2$-Hb solution (200 µg mL$^{-1}$) was irradiated with NIR laser for 2 min under different conditions (none, 0.5 W cm$^{-2}$, 1.5 W cm$^{-2}$), and the number of the produced nanovesicles with different size ranges was further investigated by NTA.

## Secretion of nanovesicles

**For TEM measurement.** 10 µL of ICG@CCM-AuNC-PO$_2$-Hb was dropped onto the copper mesh and then quickly irradiated with 808 nm laser (0.5 W cm$^{-2}$, 1.5 min) to induce the generation of membrane-derived nanovesicles. After irradiation, the sample was vacuum-dried at 30 °C before TEM measurement.

**For super resolution microscope imaging.** Preparation of imaging buffer: 2.5 mL 1 M Tris and 0.5 mL 1 M NaCl were added into 10 mL H$_2$O, then 5.5 g glucose was added into the above mixture. The resulting solution (named as Buffer B) was made up to 50 mL with DI water, filled with nitrogen for 0.5 h before use. Glucose oxidase (70 mg mL$^{-1}$, 20 µL) and Catalase (17 mg mL$^{-1}$, 5 µL) were mixed to make the GLOX solution. The imaging buffer was obtained by mixing 7 µL GLOX, 70 µL MEA (1 M), and 620 µL Buffer B for N-STORM imaging.

**Dye marking of nanoparticles.** Fluorescent dye with high flicker characteristics was used to probe the nanoparticles for super-resolution microscopy. Firstly, CD44 Polyclonal Antibody (Proteintech, 15675-1-AP, 1:200) was incubated with ICG@CCM-AuNC-PO$_2$-Hb at 4 °C overnight due to overexpressed CD44 protein from the coated cell membrane. Then the mixture was centrifuged (860 × g) for 8 min. Donkey Anti-Rabbit IgG H&L Alexa 647 (Abcam, ab150075, 1:500) was further used to combine with the modified CD44 antibody at 37 °C for 2 h and then centrifuged (860 × g) for 8 min to remove excess IgG H&L Alexa 647. The dye modified ICG@CCM-AuNC-PO$_2$-Hb were used for further imaging.

**Coating of glass slides.** The Poly-L-lysine (PLL) solution 0.1% (w/v) in H$_2$O was diluted to 0.01% with DI water before usage. The clean slides were covered by 2 mL PLL (0.01%) solution at 37 °C for 2 h. After that, the glasses were taken out for 10 min before used.

For N-SIM imaging, Alexa 647 labeled ICG@CCM-AuNC-PO$_2$-Hb and ICG@CCM-AuNC were dropped onto the PLL-coated glass slides respectively. After 5 min of standstill, super resolution microscope was used to monitor the change of these nanoparticles. At 10 s after recording, NIR laser (808 nm, 0.5 W cm$^{-2}$) was introduced until imaging for 2 min.

For N-STORM imaging, Alexa 647 labeled ICG@CCM-AuNC-PO$_2$-Hb were suspended in Buffer B and then dropped onto the PLL-coated glass slide. Fluorescence imaging was firstly recorded by super resolution microscope before laser irradiation. After irradiating for 2 min, fluorescence imaging was acquired again to record the changing of ICG@CCM-AuNC-PO$_2$-Hb before and after laser irradiation.

## Cell viability

4T1 cells and NIH3T3 cells were seeded onto 96-well plate with the density of $5 \times 10^3$ cells per well and the resulting cells were cultured for 24 h. The media were replaced with fresh serum-free ones containing AuNC, ICG@CCM-AuNC, AuNC-PO$_2$-Hb, CCM-AuNC-PO$_2$-Hb and ICG@CCM-AuNC-PO$_2$-Hb (with same Au concentration: 200 µg mL$^{-1}$) or PBS (0.1 M, pH = 7.4) respectively. After 6 h incubation, the cells were washed gently and then irradiated with or without 808 nm NIR

laser for 6 min (1 W cm$^{-2}$). After further incubation for 12 h, the media were tested by CCK-8 method. To calculate the cell viability, the absorbance at 450 nm was measured by microplate reader (Biotek 800 TS, USA).

Live-dead staining was also used to evaluate the in vitro antitumor efficacy of ICG@CCM-AuNC-PO$_2$-Hb. 4T1 cells were seeded onto 96-well plate and different formulations were added respectively, followed by NIR-irradiation. 4T1/NIH3T3 tumor spheres model was established for the live-dead staining as well. The treated cells/tumor spheres were stained with calcein-AM/PI for 30 min and observed on fluorescence microscope.

## Homologous target towards 4T1 cells

B16-F10 (or NIH3T3) and DiO-labelled 4T1 cells (labeling with DiO for 30 min before digestion) were co-cultured on 24-well plate with the density of $2.5 \times 10^4$ per well for 24 h. Then the medium was replaced by ICG@CCM-AuNC-PO$_2$-Hb (200 μg mL$^{-1}$ in serum-free RPMI-1640 medium mixed with DMEM medium) or serum-free RPMI-1640 medium mixed with DMEM medium (mix 1:1 by volume) for control group. After incubating for 6 h, the cells were irradiated with NIR laser (808 nm, 0.5 W cm$^{-2}$) for 2.5 min. The nucleus was then labeled with Hoechst33342 (1 μL for per well, 1 mg mL$^{-1}$) for 8 min, and all the groups were observed with fluorescence microscope after washing with PBS twice. For flow cytometry (FCM), B16-F10 (or NIH3T3) and DiO-labelled 4T1 cells (labeling with DiO for 30 min before digestion) were co-cultured on 24-well plate with the density of $2.5 \times 10^4$ per well for 24 h. Then the medium was replaced by ICG@CCM-AuNC-PO$_2$-Hb (200 μg mL$^{-1}$ in serum-free RPMI-1640 medium mixed with DMEM medium) or serum-free RPMI-1640 medium mixed with DMEM medium (mix 1:1 by volume) for control group. After incubating for 6 h, the cells were irradiated with NIR laser (808 nm, 0.5 W cm$^{-2}$) for 2.5 min. After washing with PBS twice, the cells were collected by centrifugation (100 × g, 3 min) and detected by FCM.

## Intracellular ROS detection

4T1 cells with a density of $4 \times 10^4$ cells per well were seeded in 24-well plate. After culturing for 24 h, AuNC, AuNC-PO$_2$-Hb, ICG@CCM-AuNC, CCM-AuNC-PO$_2$-Hb, ICG@CCM-AuNC-PO$_2$-Hb with same equivalent AuNC concentration (120 μg mL$^{-1}$) were added respectively. The cells were further incubated for 12 h and then washed with PBS. The cells were incubated with DCFH-DA (10 μM, for CLSM detection or flow cytometry analysis). Subsequently, the resulting cells were washed with PBS and then irradiated with NIR laser (808 nm, 1.0 W cm$^{-2}$) for 2 min. The nucleus was stained with Hoechst 33342.

## Enhanced penetration

4T1 and NIH3T3 cells (1:4) at a density of $1.5 \times 10^3$ cells per well were seeded in a 96 U-shaped well (Primesurface, Japan) and the resulting cells were cultured for 5 days for the formation of MCTs[43]. Next, the tumor spheroids were incubated with ICG@CCM-AuNC, ICG@CCM-AuNC-PO$_2$-Hb and ICG@CCM with same ICG concentration of 5 μg mL$^{-1}$ for 12 h. Then, the tumor spheroids were washed thrice with PBS and irradiated with 808 nm laser (0.5 W cm$^{-2}$, 1.5 min) for the production of nanovesicles. After 1 h incubation, the images of the tumor spheroids were acquired by tomoscan using Z-stack imaging with 20 μm intervals from the top to the middle of the spheroid by CLSM.

## In vivo temperature measurement of tumor-bearing mice

In vivo photothermal effects were also determined with orthotopic breast tumor models. The tumor-bearing mice were prepared by subcutaneous injection in the right hind limbs with 0.15 mL of cell suspension (the concentration of 4T1 cells was $1.5 \times 10^6$ per mL). The tumor-bearing mice were injected intravenously with

PBS, ICG@CCM-AuNC, AuNC-PO$_2$-Hb, CCM-AuNC-PO$_2$-Hb and ICG@CCM-AuNC-PO$_2$-Hb (with same Au concentration, 20 mg kg$^{-1}$ per mouse) respectively. After 24 h, the mice were anesthetized and the tumors were exposed to 808 nm laser at 1 W cm$^{-2}$ for 3 min, followed by thermal imaging.

## In vivo and ex vivo imaging

The tumor-bearing mice were prepared by subcutaneous injection in the right hind limbs with 0.15 mL of cell suspension (the concentration of 4T1 cells was $1.5 \times 10^6$ per mL). For in vivo imaging, 4T1 tumor-bearing BALB/c nude mice ($n = 3$) were injected intravenously with 150 μL free ICG, ICG-AuNC and ICG@CCM-AuNC-PO$_2$-Hb in PBS respectively with the same concentration of ICG (45 μg mL$^{-1}$). After 24 h, all mice were gaseous anesthetized and in vivo fluorescence imaging was recorded by small animal imaging system (excitation/ emission = 760/830 nm). Then, all mice were sacrificed and the tumors, hearts, livers, spleens, lungs, kidneys were collected for ex vivo signals acquiring.

To explore in vivo effects of irradiation on nanomachines, 4T1 tumor bearing BALB/c nude mice ($n = 3$) were injected intravenously with 150 μL ICG@CCM-AuNC-PO$_2$-Hb with the same ICG concentration (50 μg mL$^{-1}$). After 24 h of injection, all the mice were recorded for in vivo fluorescence imaging. The mice were irradiated with laser (0.5 W cm$^{-2}$) for 1.5 min immediately after imaging, followed by another fluorescence imaging at the time point of 26 h.

## In vivo tumor growth inhibition and safety analysis

The tumor-bearing mice were prepared by subcutaneous injection in the right hind limbs with 0.15 mL of cell suspension (the concentration of 4T1 cells was $1.5 \times 10^6$ per mL). BALB/c mice with subcutaneous 4T1 xenografts (grew up to approximately 200-400 mm$^3$) were divided into 6 groups randomly ($n = 5$): (1) PBS, (2) ICG@CCM-AuNC + NIR, (3) AuNC-PO$_2$-Hb + NIR, (4) CCM-AuNC-PO$_2$-Hb + NIR, (5) ICG@CCM-AuNC-PO$_2$-Hb, (6) ICG@CCM-AuNC-PO$_2$-Hb + NIR. The concentration of Au used was 20 mg kg$^{-1}$. For NIR laser irradiation of group (2) and (6), the mice were first irradiated for 1.5 min with 808 nm laser (0.5 W cm$^{-2}$) after 24 h post-injection, and then irradiated (1 W cm$^{-2}$) for 3 min after another 1 h. Group (3) and (4) were just irradiated with laser of 808 nm (1 W cm$^{-2}$) for 3 min. Tumor volumes and body weights of mice were measured every two days. After 24 days, the mice were euthanized and serum samples were collected. Biochemical parameters including aspartate aminotransferase (AST), alanine aminotransferase (ALT), Glutamyl transpeptidase (GGT), total bilirubin (TBIL), blood urea nitrogen (UREA), creatinine (CRE) were analyzed automatically using Pointcare M3. The main organs including liver, spleen, kidney, heart, and lung were collected for further hematoxylin-eosin (H&E) staining.

## Antitumor efficiency on orthotopic breast tumor model

To establish the orthotopic breast tumor model, 4T1 cells ($1.5 \times 10^6$ cells per 100 μL) were injected into the mammary fat pad beneath the second mammary gland in mice. After 10 days of tumor growth, the mice were divided into 6 groups randomly ($n = 5$): (1) PBS, (2) ICG@CCM-AuNC + NIR, (3) AuNC-PO$_2$-Hb + NIR, (4) CCM-AuNC-PO$_2$-Hb + NIR, (5) ICG@CCM-AuNC-PO$_2$-Hb, (6) ICG@CCM-AuNC-PO$_2$-Hb + NIR. The concentration of Au used was 20 mg kg$^{-1}$. For NIR laser irradiation of group (2) and (6), the mice were first irradiated for 1.5 min with 808 nm laser (0.5 W cm$^{-2}$) after 24 h post-injection, and then irradiated (1 W cm$^{-2}$) for 3 min after another 1 h. Group (3) and (4) were just irradiated with laser of 808 nm (1 W cm$^{-2}$) for 3 min. Tumor volumes and body weights of mice were measured every two days. After 16 days, the mice were euthanized and tumors were collected for weighed and photographed.

## Detection of tumor hypoxia

To detected the hypoxia status of the 4T1 tumor, mice were randomly divided into six groups: (1) PBS, (2) ICG@CCM-AuNC + NIR, (3) AuNC-PO$_2$-Hb + NIR, (4) CCM-AuNC-PO$_2$-Hb + NIR, (5) ICG@CCM-AuNC-PO$_2$-Hb, (6) ICG@CCM-AuNC-PO$_2$-Hb + NIR, $n = 3$ for each group. The concentration of Au used was 20 mg kg$^{-1}$. For NIR laser irradiation of group (2) and (6), the mice were first irradiated for 1.5 min with 808 nm laser (0.5 W cm$^{-2}$) after 24 h post-injection, and then irradiated (1 W cm$^{-2}$) for 3 min after another 1 h. Group (3) and (4) were just irradiated with laser of 808 nm (1 W cm$^{-2}$) for 3 min. The tumors were collected 24 h after last NIR irradiation for Anti-HIF-1 alpha antibody (Abcam, ab179483, 1:200) staining. After an overnight incubation, the primary antibody residue was washed with PBS, and then Goat pAb to Rb IgG (Alexa Fluor 488) (Abcam, ab150077, 1:500) was incubated for 1 h at 37 °C. The green fluorescence representing hypoxia regions.

## Hemolysis assay

EDTA-treated whole blood (1 mL) from healthy balb/c mouse was centrifuged (860 × $g$) for 8 min. The supernatant was discarded and the obtained red blood cells (RBCs) were washed three times. The purified RBCs (0.2 mL) were dispersed in 1 mL PBS to obtain the RBC solution. 20 μL RBC solution was added to 480 μL ICG@CCM-AuNC-PO$_2$-Hb (the final concentration of materials was 20 μg mL$^{-1}$, 100 μg mL$^{-1}$, 200 μg mL$^{-1}$, 500 μg mL$^{-1}$, 1000 μg mL$^{-1}$). RBC solution (20 μL) incubated with PBS (480 μL) and deionized water (480 μL) were used as a negative and positive control, respectively. The mixture was incubated at 37 °C for 4 h while shaking (10 x $g$). After the incubation period, the solution was photographed after centrifuged (860 x $g$) for 8 min and 100 μL supernatant was transferred to 96-well plate for absorbance measurement. The absorbance value of the supernatant was measured at 570 nm. The percent hemolysis was calculated using the following formula:

Hemolysis (%) = (sample absorbance-negative control absorbance)/ (positive control absorbance-negative control absorbance) × 100

## Statistical analysis

Data analysis was conducted using the software Microsoft Office Excel 2019, Graphpad Prism 7, and Image J (1.52a). The mean ± SD were determined for all the treatment groups. Statistical analysis was performed by Student's $t$-test (two-tailed). $P < 0.05$ was considered representative of a statistically significant difference between two groups.

## Reporting summary

Further information on research design is available in the Nature Portfolio Reporting Summary linked to this article.

## Data availability

The source data generated in this study are provided in the Supplementary Information/Source Data file. The full image dataset is available from the corresponding author upon request. Source data are provided with this paper. Figshare. Dataset. https://doi.org/10.6084/m9.figshare.23769450.

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

## Acknowledgements

This work was supported by the National Key Research and Development Program of China (2022YFA1206900, Y.T.) and the National Natural Science Foundation of China (Grant No. 22175083, Y.T.; 51973241, F.P.; 31900567, H.T.).

## Author contributions

J.G. and Y.T. conceived and designed the experiments. J.G. and H.Q. synthesized and characterized the ICG@CCM-AuNC-PO2-Hb. F.W. and L.L. extracted and purified the cell membranes. J.G. and H.T. carried out the in vitro and the cell experiments. H.W. helped with the tumor model in mice establishment. J.G. and H.Q. performed the in vivo animal experiments. S.W., J.O., and Y.Y. analyzed animal results. J.G. wrote the first draft of the manuscript with contributions from all co-authors. All the authors have discussed the results and approved the final version.

## Competing interests

The authors declare no competing interests.
