## [Peer Review File · Nature Communications]

Reviewers' Comments:

Reviewer #1:

Remarks to the Author:

Gao et al developed biomimetic nanodevices capable of in-situ secreting cell-membrane-derived nanovesicles with smaller sizes under near infrared laser irradiation for enhanced tumor penetration. Upon laser irradiation, the synthesized AuNCs in situ generated heat due to the surface plasmon resonance effect, thus inducing phase transition of the loaded PFO. Gasified PFO was able to cross the pores of AuNCs and extrude the coated cell membrane to form nanovesicles with smaller size, resulting in enhanced tumor penetration and synergistic photothermal and photodynamic therapy. The design of bubble nanomachine sounds very interesting and the manuscript is also well-written. Therefore, this reviewer recommends it to be accepted for publication in Nature Communications after minor revisions.

1. After cell membrane coating and Hb loading, the atomic percentage of carbon has been increased obviously (Figure 1k). However, how can the author get the conclusion that it has a 12% increasement compared with that shown in Supplementary Fig. 1g?
2. Some experimental procedures are missing. Please add entire experimental details in supporting information since they are very important for others to repeat. For example, the procedure of hemolytic experiment is missing (Supplementary Fig. 14).
3. The author should provide the loading capacity of hemoglobin in the manuscript since the loading capacity of Hb is also related to the delivery of oxygen in the whole system.
4. Although the photodynamic effect of the bubble nanomachines is very well characterized at the cellular level (Supplementary Fig. 7), I suggest the authors to provide more evidences about the ROS generation at the material level.

Reviewer #2:

Remarks to the Author:

In the present manuscript, Gao et al. reported an Au nanocage based therapeutic platform for tumor penetration therapy by in situ generation of nanobubbles. Tumor stroma has been a challenge for drug delivery. The authors try to generate small vesicles from cell membrane coated Au nanocage loaded with perfluorohexane by laser-triggered oxygen generation. The design is interesting but the presented data is insufficient to demonstrate the mechanism. More importantly, the authors failed to explain how gas generation can produce nanovesicles? Gases can cross easily and rapidly the membrane. Overall, the manuscript does not sufficiently support the hypothesis and therefore is not suitable for publication in its current form. Specific comments are as follows:

1. Cell membrane has very high permeability of gas, the authors try to generate oxygen to 'extrude' the cell membrane to produce vesicles, however the membrane bind tightly with the Au cage through positive and negative interactions, and the gas can easily cross the membrane, it would be difficult to impact the membrane and generate vesicles. Instead, I doubt if the heating may have damaged the cell membrane, leading to the fluorescence under microscope. The authors need to provide more evidence for demonstration, such as TEM images, and other gas/force induced vesicle formation test for comparison.
2. The author stated that "holes of ECM are typically smaller than 40 nm", this statement is not correct. There is no evidence showing the ECM 40 nm, cited papers are reviews without data, and the original paper (10.1073/pnas.081626898) described both narrow spacing (20–40 nm) and loosen spacing (75–130 nm). How can the authors know the 4T1 is the narrow spacing type. Be that as it may, the authors did not show the generation of nanovesicles less than 40 nm. In addition, why there are more particles for all sizes after laser irradiation.
3. For in vitro tumor spheroid penetration, the authors need to add more groups for comparison, such as AuNC without membrane coating, ICG@CCM without AuNC.
4. Figure S7a, the green fluorescence seems out of focus and the number of cell is too less for presentation.
5. For fully present the dense stroma of the tumor microenvironment, the orthotopic model is suggested.
6. The tumor site in Figure 5c did not match with Figure 5a and 5d.
7. Writing of the manuscript needs professional English editing.

Reviewer #3:

The paper has a good novelty and rich results, but there are some problems as follows.

- 1) Figure 2 g): Please add the UV-Vis absorption spectrogram of Hb.
- 2) How to show whether the enrichment of C/N/O is from the cell membrane or Hb protein? Please add the experiments that can demonstrate the successful loading of Hb.
- 3) In this paper, the distribution of nanosystems at tumor sites in vivo is demonstrated by in vivo fluorescence experiments, and the authors write at the end of Main section that the photothermal effect leads to the formation of smaller sized cell membrane-derived nanovesicles that further promote effective tumor penetration. Therefore please add the in vivo distribution map of ICG@CCM-AuNC-PO2-Hb after laser irradiation to demonstrate this effect.
- 4) The dosing concentration in the animal experiments in the method section is 20mg/kg based on Au, is the dosing concentration too high, what is the proportion of Au in the whole nano system, and what is the loading capacity of ICG?
- 5) The two groups of preparations examined were ICG@CCM-AuNC-PO2-Hb and ICG@CCM-AuNC at a concentration of 1 mg/mL, what substance concentration does this concentration refer to specifically?
- 6) The stability of hemoglobin is poor, how did the authors ensure its stability throughout the preparation process?
- 7) What is the Ag-AuNC in Figure 2 b), please write it clearly.
- 8) The two transmission electron micrographs of Figure 2 c) and e) do not seem to be significantly different, where the cell membrane structure is shown? Figure 2f is a magnification of Figure 2e, and Figure 2 c magnification would have a similar effect.
- 9) Line 170: "Only 4 ° C decline was observed after four heating-cooling cycles, indicating superb photostability of our ICG@CCM-AuNC-PO2-Hb ". Many photothermal nanoparticles reported in the literature maintain their temperature after repeated laser irradiation, can a 4 ° C decline indicate superb photothermal stability?
- 11) Figure 4 b): Under NIR, the cell viability of CCM-AuNC-PO2-Hb is higher than that of AuNC vs. AuNC-PO2-Hb, why? Is there a significant difference between the cell viability of ICG@CCM-AuNC and ICG@CCM-AuNC-PO2-Hb groups? If there is no significant difference, what is the significance of increasing the oxygen concentration? What is the "n" for the cell activity experiment?
- 12) What is the composition of the smaller size cell membrane-derived nanovesicles formed as a result of the photothermal effect? What is its particle size distribution? What is the effect of the photothermal effect on the original nanoparticles after the generation of nanovesicles?
- (13) During the "Measurement of the Released O₂" experiment, should the amount of O₂ released at a laser power of 1 W/cm² be supplemented? This power is used as the used power in the subsequent cell and animal experiments and should rightly be examined.

- 14) In vivo tumor suppression experiments, the reason for the difference between different groups should be further analyzed.
- 15) In many experiments containing laser irradiation, the power of the laser used and the duration of irradiation are different. Why? Please give the reasons why different power of laser and irradiation time are needed.
- 16) Figures 4a and b can be combined into one graph with statistical analysis and explanation of the differences.
- 17) What is the photothermal conversion rate of ICG@CCM-AuNC-PO2-Hb?
- 18) It is mentioned several times in the manuscript that ICG@CCM-AuNC-PO2-Hb can achieve photothermal/photodynamic therapy, but there are few experiments related to photodynamic therapy, and the corresponding in vivo and ex vivo experiments should be added.
- 19) Should the biocompatibility of ICG@CCM-AuNC-PO2-Hb on normal cells be examined?
- 20) The expression of laser density is 0.5 W in some places (line 239) and 0.5 W cm⁻² in some places (line 244), please check the whole text and revise it.

Dear Referees,

We have thoroughly revised the manuscript according to the suggestions made by the referees. The changes are marked in red in the revised manuscript. A point-by-point response to referees' comments and concerns is also provided below.

Reviewer #1 (Remarks to the Author):

Gao et al developed biomimetic nanodevices capable of in-situ secreting cell-membrane-derived nanovesicles with smaller sizes under near infrared laser irradiation for enhanced tumor penetration. Upon laser irradiation, the synthesized AuNCs in situ generated heat due to the surface plasmon resonance effect, thus inducing phase transition of the loaded PFO. Gasified PFO was able to cross the pores of AuNCs and extrude the coated cell membrane to form nanovesicles with smaller size, resulting in enhanced tumor penetration and synergistic photothermal and photodynamic therapy. The design of bubble nanomachine sounds very interesting and the manuscript is also well-written. Therefore, this reviewer recommends it to be accepted for publication in Nature Communications after minor revisions.

Answer: We are pleased to see this referee finds the manuscript interesting. As summarized by the referee, we designed a bubble machine with relatively innovative properties that can generate nanobubbles in response to photothermal stimulation for deep tumor penetration. We have also carefully modified the manuscript according to the referee's valuable suggestions.

1) After cell membrane coating and Hb loading, the atomic percentage of carbon has been increased obviously (Figure 1k). However, how can the author get the conclusion that it has a 12% increasement compared with that shown in Supplementary Fig. 1g?

Answer: We would like to thank the referee for the careful observation. We forgot to add the data of atomic percentage of AuNC group in the supporting information. The atomic percentage of AuNC group measured by EDX mapping was provided as Fig.S1g, h. As shown in Fig.S1g, it had a 12% increasement in C after coating of the surrounding organic layer and Hb loading.

2) Some experimental procedures are missing. Please add entire experimental details in

supporting information since they are very important for others to repeat. For example, the procedure of hemolytic experiment is missing (Supplementary Fig. 14).

Answer: Thank you very much for pointing this out. The detail of the “Hemolysis assay” was added into the revised supporting information. We also checked the supporting information, and all the experimental procedures should be complete now.

3) The author should provide the loading capacity of hemoglobin in the manuscript since the loading capacity of Hb is also related to the delivery of oxygen in the whole system.

Answer: Thank you for good suggestion. The loading capacity of Hb was calculated according to the relative absorbance intensity of Hb in UV-vis-NIR spectrum. Based on the standard curve of Hb shown in Fig. S3a, the loading content of Hb was calculated to $6.03 \pm 0.89\%$, which was added in the revised manuscript.

4) Although the photodynamic effect of the bubble nanomachines is very well characterized at the cellular level (Supplementary Fig. 7), I suggest the authors to provide more evidences about the ROS generation at the material level.

Answer: Thank you for your comment. SOSG fluorescent probes were further used to assess the ability of $^1\text{O}_2$ generation of different formulations under NIR irradiation. As shown in Fig.S6, the fluorescence intensity of ICG@CCM-AuNC-PO₂-Hb was 2.5 folds higher than that in blank groups, and 1.2 folds higher than that in ICG@CCM-AuNC, which was attributed to the generation of $^1\text{O}_2$ promoted by the O₂ released from ICG@CCM-AuNC-PO₂-Hb.

Reviewer #2 (Remarks to the Author):

In the present manuscript, Gao et al. reported an Au nanocage based therapeutic platform for tumor penetration therapy by in situ generation of nanobubbles. Tumor stroma has been a challenge for drug delivery. The authors try to generate small vesicles from cell membrane coated Au nanocage loaded with perfluorohexane by laser-triggered oxygen generation. The design is interesting but the presented data is insufficient to demonstrate the mechanism. More importantly, the authors failed to explain how gas generation can produce nanovesicles? Gases can cross easily and rapidly the membrane. Overall, the manuscript does not sufficiently support the

hypothesis and therefore is not suitable for publication in its current form. Specific comments are as follows:

Answer: We are pleased that this reviewer also found our work interesting. In addition to the fluorescence images and videos already presented by super-resolution microscopy, we have performed additional TEM measurement (Fig.3h) to demonstrate the nanovesicles generation upon laser irradiation. Reading through the literature, we found that the composition of cancer cell membranes could be remodeled, for example, phospholipids and cholesterol are increased in breast cancer cells, which decreases the oxygen transmission and therefore this gives the possibility of oxygen extrusion the cloaked cell membrane into the vesicles. Additional experiments and responses have been added as followed.

1) Cell membrane has very high permeability of gas, the authors try to generate oxygen to 'extrude' the cell membrane to produce vesicles, however the membrane bind tightly with the Au cage through positive and negative interactions, and the gas can easily cross the membrane, it would be difficult to impact the membrane and generate vesicles. Instead, I doubt if the heating may have damaged the cell membrane, leading to the fluorescence under microscope. The authors need to provide more evidence for demonstration, such as TEM images, and other gas/force induced vesicle formation test for comparison.

Answer: Thank you for your concern. Actually, there are quite some differences between the normal cell membrane and tumor cell membrane. When compared with normal cells, the membrane of tumor cells was normally remodeled, especially in breast cancer cells (the membrane phospholipids with most increased), as described in the previous report "...The membrane phospholipids including phosphatidylcholines (PC), PEs, and PIs, as well as sphingomyelins (SM) and ceramides (Cer), were the most increased lipids in tumors..." (DOI: 10.1158/0008-5472.CAN-10-3894). It was also reported that through over-activating their endogenous synthesis, excessive lipids and cholesterol in cancer cells would be stored in lipid droplets of cell membrane (DOI:10.1038/onc.2015.49). As the cholesterol content increases, the physical properties of cell membrane would be influenced, which may generate a physical barrier to oxygen diffusion. The results were ascribed to the fact that "...cholesterol reduces the overall solubility of oxygen in the bilayer, possibly diminishing the volume of oxygen that can be transmitted laterally within the membrane..."

(10.1016/j.bpj.2017.04.046). Therefore, we believed that the generated O₂ is possible to extrude the tumor cell membrane to produce vesicles. As suggested, TEM was further used to capture the vesicle formation. As shown in Fig.3h, membrane-derived vesicles were observed on the surface of AuNC after laser irradiation.

2) The author stated that “holes of ECM are typically smaller than 40 nm”, this statement is not correct. There is no evidence showing the ECM 40 nm, cited papers are reviews without data, and the original paper (10.1073/pnas.081626898) described both narrow spacing (20–40 nm) and loosen spacing (75–130 nm). How can the authors know the 4T1 is the narrow spacing type. Be that as it may, the authors did not show the generation of nanovesicles less than 40 nm. In addition, why there are more particles for all sizes after laser irradiation.

Answer: Thank you for your concerns. The above-mentioned statement was corrected in the revised manuscript. Actually, we did not mean that our bubble nanomachine can accurately produce vesicles smaller than 40 nm for penetrating the holes of ECM. In our design, the main function of the bubble nanomachine is to target the homotypic tumor, and then generate the hyperthermia-induced vesicles smaller than the mother to penetrate deeply into the core of tumor. In our experiments, the generated nanovesicles were widely distributed as shown in the TEM images (Fig.3h), and its hydrodynamic size was ranging from 43 nm to 120 nm (Fig.S8). Therefore, the increase in number of nanoparticles for all sizes after irradiation was observed.

3) For in vitro tumor spheroid penetration, the authors need to add more groups for comparison, such as AuNC without membrane coating, ICG@CCM without AuNC.

Answer: Thank you for your suggestion. Due to the hard loading of fluorescent ICG in AuNC without membrane coating, we conducted the in vitro tumor spheroid penetration of ICG@CCM. Tumor spheroids (n=3) were incubated with ICG@CCM with ICG concentration of 5 ug mL⁻¹ for 12 h, followed by same NIR irradiation condition. And the data were collected by confocal microscope with the same parameters (Figure 4e). When compared with our ICG@CCM-AuNC-PO₂-Hb, ICG@CCM can penetrate somehow, but most of them just remained on the surface of the tumor.

4) Figure S7a, the green fluorescence seems out of focus and the number of cell is too

less for presentation.

Answer: Thank you for your careful observation. The intracellular ROS detection was performed again in order to present the better results. As shown in new Fig.S13a, the group of ICG@CCM-AuNC-PO₂-Hb induced more ROS production under the same condition of NIR irradiation.

5) For fully present the dense stroma of the tumor microenvironment, the orthotopic model is suggested.

Answer: Thank you for nice suggestion. In the revised manuscript, orthotopic breast cancer model was further established to evaluate our bubble nanomachines. Similar results were obtained, which can be seen as Fig.S18 in the revised supporting information.

6) The tumor site in Figure 5c did not match with Figure 5a and 5d.

Answer: Thank you for careful observation. In Fig.5a, nude mice were injected with cancer cells in the left site, which was better for the observation of *in vivo* distribution. While for Fig.5c, cancer cells were injected on the right side of the BALB/c mice.

7) Writing of the manuscript needs professional English editing.

Answer: Thank you for your suggestion. The language of our manuscript has been polished.

Reviewer #3 (Remarks to the Author):

The paper has a good novelty and rich results, but there are some problems as follows.

Answer: We are pleased to see that the referee evaluated this manuscript for “good novelty and rich results”. This referee also finds some points of improvement for the manuscript. For this we would like to thank him for his suggestions. According to the referee’s concerns, detailed responses are given below.

1) Figure 2 g): Please add the UV-Vis absorption spectrogram of Hb.

Answer: We want to thank the referee for nice suggestion. The UV-Vis absorption spectrogram of Hb was measured and added in the new Fig.2g. The absorption peak of Hb was 405 nm, which was consistent with previous reports.

2) How to show whether the enrichment of C/N/O is from the cell membrane or Hb protein? Please add the experiments that can demonstrate the successful loading of Hb.

Answer: Thank you for your concerns. In order to compare the difference between AuNC and ICG@CCM-AuCN-PO₂-Hb, elemental mapping of AuNC was conducted and 12% C enrichment was clearly observed after successful Hb loading and cell membrane coating (Fig.S1g). Moreover, SDS protein analysis of different preparations was further measured, as shown in Fig.S2. The protein bands of around 25 kD (hemoglobin-specific) can be observed clearly, indicating successful loading of Hb.

3) In this paper, the distribution of nanosystems at tumor sites *in vivo* is demonstrated by *in vivo* fluorescence experiments, and the authors write at the end of Main section that the photothermal effect leads to the formation of smaller sized cell membrane-derived nanovesicles that further promote effective tumor penetration. Therefore please add the *in vivo* distribution map of ICG@CCM-AuCN-PO₂-Hb after laser irradiation to demonstrate this effect.

Answer: Thank you for your suggestion. 4T1 tumor bearing BALB/c nude mice (n = 3) were injected intravenously with 150 μ L ICG@CCM-AuNC-PO₂-Hb (with the same ICG concentration of 50 μ g/mL). After 24 h of injection, all the mice were recorded by *in vivo* fluorescence imaging. The mice were then irradiated with laser for 1.5 min immediately after imaging, followed by another fluorescence imaging at 26 h. As shown in Fig. S15, the increase in fluorescence intensity was clearly observed in tumor after laser irradiation. It is another evidence to prove that photothermal effect can induce the formation of smaller sized nanovesicles to promote effective tumor penetration.

4) The dosing concentration in the animal experiments in the method section is 20 mg/kg based on Au, is the dosing concentration too high, what is the proportion of Au in the whole nano system, and what is the loading capacity of ICG?

Answer: Thank you for your concern. In our opinions, “20 mg/kg based on Au” is an appropriate dose for administration, and we normally injected 400 μ g materials based on Au into mice (20 g per mouse), which was comparable with previous report. “...intravenously injected with saline, DOX, CAuNs, CDAuNs, CAuNs with NIR laser and CDAuNs with NIR laser irradiation at 2.5 W cm⁻² for 5 min (5 mg/kg of DOX and 91 mg/kg of CAuNs) every 3 days for 4 times, respectively (DOI:

10.1002/adfm.201604300)". ICP-AES and fluorescence spectroscopy were further used to quantify the proportion of Au, and the content of Au in the whole nanomachine was $67.1 \pm 12\%$, and the loading capacity of ICG was calculated to $11.19 \pm 1.8\%$ based on the standard curve of ICG (Fig.S3).

5) The two groups of preparations examined were ICG@CCM-AuNC-PO₂-Hb and ICG@CCM-AuNC at a concentration of 1 mg/mL, what substance concentration does this concentration refer to specifically?

Answer: Thank you for your concerns. 1 mg/mL here refers to the concentration of gold as the equivalent.

6) The stability of hemoglobin is poor, how did the authors ensure its stability throughout the preparation process?

Answer: Thank you for your concern. In order to maintain the Hb activity, Hb loading and subsequent membrane extrusion were always kept at low temperature (ice bath). To verify the stability of the loaded Hb, ICG@CCM-AuNC-PO₂-Hb were shaken for 24 h at 37 °C and the released Hb were collect for far-Uv CD spectra testing (Fig.S4a). Secondary structure was calculated based on the raw CD data summarized in Fig.S4b. Negligible change in the secondary structure of Hb was observed, suggesting the good stability of Hb throughout the preparation process.

7) What is the Ag-AuNC in Figure 2 b), please write it clearly.

Answer: Thank you very much for pointing this out. More description of Ag-AuNC was added in the revised manuscript. "After adding increasing amounts of H₂AuCl₄ solution, the resulting AgNC were gradually transformed into Ag-AuNC due to galvanic replacement reaction,³² including the initiation of partially hollow nanostructure composed of Ag/Au alloy and final formation of AuNC with pores in the walls (Fig.2b-c and Supplementary Fig.S1b-c)."

8) The two transmission electron micrographs of Figure 2 c) and e) do not seem to be significantly different, where the cell membrane structure is shown? Figure 2f is a magnification of Figure 2e, and Figure 2 c magnification would have a similar effect.

Answer: Thank you for your concerns. Because the contract between cell membrane and gold is quite different, therefore the coated cell membrane is hard to be observed

clearly in the TEM. New images of ICG@CCM-AuNC-PO₂-Hb were added to replace the old ones in Fig. 2e-f, which relatively displayed the outer cell membrane coating.

9) Line 170: "Only 4 °C decline was observed after four heating-cooling cycles, indicating superb photostability of our ICG@CCM-AuNC-PO₂-Hb". Many photothermal nanoparticles reported in the literature maintain their temperature after repeated laser irradiation, can a 4 °C decline indicate superb photothermal stability?

Answer: Thank you for your concerns. The description was modified as followed: "Only 4 °C decline was observed after four heating-cooling cycles, indicating relative photostability of our ICG@CCM-AuNC-PO₂-Hb".

11) Figure 4 b): Under NIR, the cell viability of CCM-AuNC-PO₂-Hb is higher than that of AuNC vs. AuNC-PO₂-Hb, why? Is there a significant difference between the cell viability of ICG@CCM-AuNC and ICG@CCM-AuNC-PO₂-Hb groups? If there is no significant difference, what is the significance of increasing the oxygen concentration? What is the "n" for the cell activity experiment?

Answer: Thank you for your concerns. We guess that the residual of nanoparticles might affect the Uv-vis absorption at 570 nm during MTT. Therefore, CCK-8 assay was used to determine the dark viability and phototoxicity of 4T1 cells again. As shown in the new Fig. 4a, the cell phototoxicity of CCM-AuNC-PO₂-Hb (53 %) were lower than that of AuNC (90 %) and AuNC-PO₂-Hb (66.7%), which indicated the better efficacy of cell membrane coating strategy.

After statistical analysis, we found a statistical significance (**P=0.0075) between ICG@CCM-AuNC and ICG@CCM-AuNC-PO₂-Hb groups under irradiation. Compared with the ICG@CCM-AuNC (25.4%), the viability of ICG@CCM-AuNC-PO₂-Hb was only decreased by 1.5%, because the NIR-triggered penetration effect of ICG@CCM-AuNC-PO₂-Hb on 2D cell level is difficult to reflect. Therefore, the two groups showed similar therapeutic effects in cell viability experiments. However, when treated with animal models, ICG@CCM-AuNC-PO₂-Hb group had significantly inhibitory effect on tumor tissue more than the group of ICG@CCM-AuNC, which was attributed to its hyperthermia-triggered bubbles for deep treatment (Fig. 5d-f). In the cell activity experiment, data are the representative of five independent experimental holes. (n=5).

12) What is the composition of the smaller size cell membrane-derived nanovesicles formed as a result of the photothermal effect? What is its particle size distribution? What is the effect of the photothermal effect on the original nanoparticles after the generation of nanovesicles?

Answer: Thank you for your concerns. The main composition of the cell membrane-derived nanovesicles was ICG-anchored 4T1 cell membrane and hemoglobin, which can be proved by the SDS-PAGE protein analysis in Fig. S2. The particles size distribution of ICG@CCM-AuNC-PO₂-Hb after NIR irradiation was displayed in the Fig 3 d-g. The size distribution of the hyperthermia-triggered nanovesicles were tested by DLS, and the nanoparticle size range was 43-120 nm (Fig.S8). After the hyperthermic stimulation upon the nanomachine, the composition of the outer cell membrane would be correspondingly reduced. As the TEM images shown in the figure 2 e-f and figure 3 h, the cell membrane would disappear or the thickness was decreased after the nanovesicles generation.

13) During the "Measurement of the Released O₂" experiment, should the amount of O₂ released at a laser power of 1 W/cm² be supplemented? This power is used as the used power in the subsequent cell and animal experiments and should rightly be examined.

Answer: Thank you for your careful observation. In the "Measurement of the Released O₂" experiment, we actually used the 808 nm laser at the power of 1 W cm⁻². As the caption shown in the Fig. 3c, "O₂ concentration after addition of ICG@CCM-AuNC or ICG@CCM-AuNC-PO₂-Hb into the deoxygenated water with or without NIR irradiation (808 nm, 1 W cm⁻²)." was the correct description. Then the corresponding part about experiment method was also corrected in the "Measurement of the Released O₂" of revised Supporting Information.

14) In vivo tumor suppression experiments, the reason for the difference between different groups should be further analyzed.

Answer: Thank you for your suggestion. We added corresponding explanations for differences between groups based on the statistical analysis. The group of ICG@CCM-AuNC-PO₂-Hb without laser irradiation hardly showed a tumor inhibitory effect because of absent photothermal or photodynamic therapy. While certain antitumor effect was observed on the groups under irradiation treated with ICG@CCM-AuNC

(53%) or CCM-AuNC-PO₂-Hb (63%) when compared with that of PBS group. Among that, due to the generated ROS for combinational therapy, the group of ICG@CCM-AuNC was superior to the CCM-AuNC-PO₂-Hb treatment, but it was less effective than the group of ICG@CCM-AuNC-PO₂-Hb because no nanovesicles were produced for deep therapy.

15) In many experiments containing laser irradiation, the power of the laser used and the duration of irradiation are different. Why? Please give the reasons why different power of laser and irradiation time are needed.

Answer: Thank you for your suggestion. There are two different powers of laser irradiation namely 0.5 W cm⁻² and 1 W cm⁻². 0.5 W cm⁻² with short time irradiation was used for the formation of membrane-induced nanovesicles under lower temperature, while 1.0 W cm⁻² with long time irradiation was used for photothermal (AuNC) /photodynamic (ICG) therapy to suppress tumor.

16) Figures 4a and b can be combined into one graph with statistical analysis and explanation of the differences.

Answer: Thank you for your suggestion. The dark viability and phototoxicity of 4T1 cells were measured again with CCK-8 and they were combined into one graph for better comparison as new Fig. 4a. The explanation of the differences was added in the revised manuscript.

17) What is the photothermal conversion rate of ICG@CCM-AuNC-PO₂-Hb?

Answer: Thank you for your concerns. According to D. K. Roper's methods, the linear time data versus $-\ln\theta$ was obtained from the ICG@CCM-AuNC-PO₂-Hb cooling period after NIR heating (Fig. S5a-b). Inputting values for m_i (0.25 g) and C_p (4.2 J g⁻¹ °C⁻¹) the photothermal conversion efficiency was calculated to be 26.81 ± 1.3%.

18) It is mentioned several times in the manuscript that ICG@CCM-AuNC-PO₂-Hb can achieve photothermal/photodynamic therapy, but there are few experiments related to photodynamic therapy, and the corresponding in vivo and ex vivo experiments should be added.

Answer: Thank you for your suggestion. At the cellular level, the photodynamic effect of our ICG@CCM-AuNC-PO₂-Hb was characterized by fluorescence probe DCFH-

DA, indicating powerful ROS generation of bubble nanomachine according to fluorescence imaging (Fig. S13a) and flow cytometry (Fig. S13b-c). For *in vivo* experiments, the photodynamic effect also can be proved in the tumor growth inhibition of mice models. As shown in Fig. 5f, the combination of ICG@CCM-AuNC-PO₂-Hb with laser irradiation displayed the best antitumor efficacy and the tumor weight was decreased considerably by 95% when compared with that of ICG@CCM-AuNC group, which was another evidence for photodynamic activity of our nanomachine. Furthermore, hypoxia-inducible factor (HIF)-1 α immunofluorescence staining of tumor sections was added to evaluate the *in vivo* hypoxic condition after PDT. As displayed in Fig S19, stronger green fluorescence was detected on the group treated with ICG@CCM-AuNC-PO₂-Hb under irradiation when compared with that of group without NIR. Because of the aggravated oxygen consumption after PDT, the group of ICG@CCM-AuNC without O₂ delivery showed the highest expression of HIF-1 α .

19) Should the biocompatibility of ICG@CCM-AuNC-PO₂-Hb on normal cells be examined?

Answer: Thank you for your suggestion. 3T3 cell lines were used to measure the dark viability and phototoxicity of ICG@CCM-AuNC-PO₂-Hb, which was added as Fig. S9. When 3T3 cells were treated with the same formulations, their cytotoxicity was generally lower than that in the groups of 4T1 cells, indicating the good biocompatibility of our nanodevices.

20) The expression of laser density is 0.5 W in some places (line 239) and 0.5 W cm⁻² in some places (line 244), please check the whole text and revise it.

Answer: Thank you for your careful observation. The corresponding sentence was corrected: “N-STORM images of ICG@CCM-AuNC-PO₂-Hb obtained by super-resolution fluorescence microscopy before and after NIR irradiation (0.5 W cm⁻², 5 min), scale bar =200 nm.” The manuscript was also thoroughly checked.

With these modifications, we hope now the manuscript is acceptable for publication in Nature Communications.

Sincerely yours,
Yingfeng

Reviewers' Comments:

Reviewer #1:

Remarks to the Author:

It can be accepted now

Reviewer #2:

Remarks to the Author:

The authors have adequately addressed my comments.

Reviewer #3:

Remarks to the Author:

The author has already answered my question and I have no new questions.

Dear referees,

We are happy to see that all the referees agree for publication of our manuscript in Nature Communications.

Best regards,

Yingfeng